# Activation of AMPD2 drives metabolic dysregulation and liver disease in mice with hereditary fructose intolerance
Ana Andres-Hernando[1], David J. Orlicky [2], Masanari Kuwabara [3,4], Mehdi A. Fini[5], Dean R. Tolan [6], Richard J. Johnson[7] & Miguel A. Lanaspa [1] ✉

Hereditary fructose intolerance (HFI) is a painful and potentially lethal genetic disease caused by a mutation in aldolase B resulting in accumulation of fructose-1-phosphate (F1P). No cure exists for HFI and treatment is limited to avoid exposure to fructose and sugar. Using aldolase B deficient mice, here we identify a yet unrecognized metabolic event activated in HFI and associated with the progression of the disease. Besides the accumulation of F1P, here we show that the activation of the purine degradation pathway is a common feature in aldolase B deficient mice exposed to fructose. The purine degradation pathway is a metabolic route initiated by adenosine monophosphate deaminase 2 (AMPD2) that regulates overall energy balance. We demonstrate that very low amounts of fructose are sufficient to activate AMPD2 in these mice via a phosphate trap. While blocking AMPD2 do not impact F1P accumulation and the risk of hypoglycemia, its deletion in hepatocytes markedly improves the metabolic dysregulation induced by fructose and corrects fat and glycogen storage while significantly increasing the voluntary tolerance of these mice to fructose. In summary, we provide evidence for a critical pathway activated in HFI that could be targeted to improve the metabolic consequences associated with fructose consumption.

Intake of sugar has exponentially increased from 35–45 to 70–80 grams/day after the introduction of high fructose corn syrup in the 1970s[1,2]. it is very difficult to avoid exposure to sugar in today's culture despite efforts from the American Heart Association and the World Health Organization recommending the reduction of its intake as it unequivocally increases the risk for dental caries and metabolic syndrome[3].

One group that suffers from the widespread use of added sugars are individuals with Hereditary Fructose Intolerance (HFI). HFI is an autosomal recessive disease in the *aldob* gene with an incidence of 1:20,000[4,5] although the frequency is likely higher due to lack of diagnosis[6]. Aldolase B, the protein codified by the *aldob* gene, mediates the second step in fructose metabolism and converts fructose-1-phosphate (F1P) to glyceraldehyde and dihydroxyacetone phosphate. HFI is characterized by inefficient fructose metabolism and the accumulation of F1P. Subjects with HFI develop severe reactions following fructose ingestion, with abdominal pain, vomiting, diarrhea, symptomatic hypoglycemia, hyperuricemia, and other pathologies[6,7]. One of the greatest risk periods for subjects with HFI is for the newborn infant being weaned from breast milk, since the autosomal recessive nature of the disease is such that the parents are often unaware of possessing the aldoB- allele. Acute exposure to fructose in subjects with non-functional aldolase B may result in acute hypoglycemia, seizures, lactic acidosis, coma or death[7–12]. Other common adverse consequences associated with HFI include chronic kidney disease and nonalcoholic steatohepatitis (NASH). Indeed, many cases of undiagnosed liver failure in infancy are due to HFI. In general, individuals with HFI develop a permanent and powerful protective aversion to sweet-tasting foods[11]. However, subjects commonly develop cirrhosis or chronic kidney disease despite attempted restriction of fructose intake[13,14]. This is likely due to the presence of small amounts and traces of added sugars in foods[15–18]. Further, our group has demonstrated that fructose is endogenously produced in the body from glucose[19] and that the metabolism of this endogenous fructose in aldolase B deficient mice can drive the disease chronically in the absence of dietary fructose[20].

[1]Division of Endocrinology, Metabolism and Diabetes, University of Colorado Denver, Aurora, CO, USA. [2]Department of Pathology, University of Colorado School of Medicine, Aurora, CO, USA. [3]Department of Cardiology, Toranomon Hospital, Tokyo, Japan. [4]Division of Public Health, Center for Community Medicine, Jichi Medical University, Tochigi, Japan. [5]Division of Pulmonary and Critical Care Medicine, University of Colorado Anschutz Medical Campus, Aurora, CO, USA. [6]Department of Biology, Boston University, Boston, MA, USA. [7]Division of Renal Diseases and Hypertension, University of Colorado Denver, Aurora, CO, USA. ✉ e-mail: Miguel.lanaspagarcia@cuanschutz.edu

In general, little is known about the sub-clinical and/or long-term effects from loss of aldolase B activity, even without fructose ingestion, or about the effects in heterozygotes who it is predicted to comprise ~2% worldwide[21]. In this regard, Debray et al, recently showed that heterozygotes may be at greater risk of developing hyperuricemia and reduced insulin sensitivity suggestive of the high susceptibility to fructose conferred by the partial loss of AldoB[22].

Aldob knockout mice (*aldob*(-/-)) phenocopy human HFI[23,24]. Here, we have aimed to identify pathways that could be targeted to ameliorate the pathologies associated with aldolase B deficiency. In particular, we have focused on AMPD2 and the purine degradation pathway, a metabolic route involved in overall energy control and its regulation[25–27] and determined whether its specific activation in the liver by fructose is an important deleterious step in the pathogenesis of HFI. We provide biochemical, histological and dietary evidence that the activation of AMPD2 and the purine degradation pathway is an important deleterious step in the pathogenesis of HFI in mice.

## Results

### Metabolic consequences associated with hyperactivation of AMPD2 by fructose in aldob(-/-) mice

The majority of dietary fructose is metabolized in the liver and gut[28–30]. Unlike wild type mice, the metabolism of fructose in *aldob*(-/-) is incomplete and results in the accumulation of fructose as fructose-1-phosphate (F1P). Phosphate is a known AMPD inhibitor[25,31]. Therefore, it is likely that the sequestration of phosphate in the F1P molecule leads to low levels of intracellular free phosphate and thus, the activation of AMPD2 in *aldob*(-/-) mice[32] (Fig. 1a). AMPD2 expression is not significantly different between wild type and *aldob*(-/-) mice (Fig. 1b). However, intracellular phosphate levels are significantly lower both at baseline ($69.93 \pm 1.73$ vs $39.42 \pm 6.55$ nmol/mg $P < 0.01$) and upon fructose exposure ($47.27 \pm 5.71$ vs $19.82 \pm 7.31$ nmol/mg $P < 0.01$, Fig. 1c) in *aldob*(-/-) mice compared to wild type counterparts. In turn, AMPD activity was found to be significantly higher in *aldob*(-/-) mice compared to wild type counterparts at baseline ($8.4 \pm 2.2$ vs $17.9 \pm 3.1$ μmol ammonia/min/mg, $P < 0.01$) and further exacerbated in mice fed a diet containing fructose ($12.6 \pm 2.6$ vs $41.8 \pm 7.5$ μmol ammonia/min/mg, $P < 0.01$, Fig. 1d) for 2 weeks (1% w/w).

At 2 weeks of fructose-exposure, higher AMPD activity in fructose-exposed *aldob*(-/-) mice was associated with a decrease in cellular energy state characterized by a low nucleotide pool (ATP, ADP and AMP, Fig. 1e) and energy charge (Fig. 1f). However, because of the rapid deamination of AMP by AMPD2, the AMP/ATP ratio is not elevated in fructose-fed *aldob*(-/-) mice (Fig. 1g) and, paradoxically, the low energy state in *aldob*(-/-) mice is not corresponded by the activation of the cell energy sensor AMPK as demonstrated by minimal phosphorylation at threonine 172 (Fig. 1h). When activated, AMPK promotes rapid lipid and glycogen mobilization to restore ATP during low energy conditions. This is mediated by the phosphorylation and inactivation of target proteins including glycogen synthase (GS) and acetyl-CoA carboxylase (ACC). As shown in Fig. 1i, GS phosphorylation at serine 641 is markedly lower in fructose-fed *aldob*(-/-) mice (Fig. 1l) resulting in higher glycogen content. Similarly, phosphorylation of ACC at serine residue 79 is minimal in fructose-fed *aldob*(-/-) mice (Fig. 1j) in association with inflammation and triglyceride accumulation (Fig. 1k, l) with large macrosteatotic areas surrounding the central vein and portal triad, and ductal reaction in the periportal area (Fig. 1k).

### AMPD2 deletion ameliorates metabolic dysregulation in aldob(-/-) mice

F1P is a metabolically active metabolite in liver and gut that increases nutrient absorption and promotes de novo lipogenesis and cell growth[33,34]. Based on this observation, we hypothesized that the metabolic imbalance in fructose-exposed *aldob*(-/-) mice could be mediated by either an F1P-specific signaling mechanism or the activation of the side-chain pathway initiated by AMPD2. To separate the effects of F1P accumulation from the activation of AMPD2, we silenced AMPD2 expression in *aldob*(-/-) mice.

Efficient silencing of AMPD2 is shown in Fig. 2a. Functional deletion of AMPD2 is demonstrated by a stepwise (wild type vs *ampd2*(+/-) vs *ampd2*(-/-)) reduction in inosine monophosphate (IMP) (Fig. 2b), and the elevation of the AMP/IMP ratio (Fig. 2c). Of interest, baseline effect of AMPD2 deletion in *aldob*(-/-) mice is also observed for other metabolic markers including hepatic energy charge (Fig. 2d) and the AMP to ATP ratio (Fig. 2e). Furthermore, and consistent with a higher AMP/ATP ratio, AMPK is hyperactivated in *aldob/ampd2*(-/-) mice (Fig. 2f). Fructose-free aldob(-/-) mice demonstrated higher baseline liver triglyceride ($1.2 \pm 0.3$ g/g in *aldob*(-/-) mice vs $0.3 \pm 0.2$ in wildtype and $0.5 \pm 0.2$ in *aldob/ampd2*(-/-) mice) and glycogen ($6.2 \pm 0.9$ g/g in *aldob*(-/-) mice vs $2.3 \pm 0.6$ in wildtype and $2.5 \pm 0.4$ in *aldob/ampd2*(-/-) mice) than wild type and *aldob/ampd2*(-/-) mice. These baseline differences are likely secondary to endogenous fructose production and metabolism from the dextrose/corn starch rich chow that mice are fed. Exposure to 1% fructose (w/w), resulted in a marked exacerbation of the metabolic phenotype (Fig. 2g–n) which was found to be dependent on the AMPD2. In this regard, fructose-exposed *aldob/ampd2*(-/-) mice demonstrated reduced liver glycogen (Fig. 2g-h), lipid accumulation (Fig. 2i, j), inflammation (Fig. 2i, j), fibrosis (Fig. 2k–n) and injury (denoted by the injury score and plasma levels of transaminases, Fig. 2 m,n), and therefore a marked protection against the characteristic steatohepatitis and liver pathology in fructose-exposed *aldob*(-/-) mice[23,24].

### Effects of AMPD2 deletion on fructose tolerance in aldob(-/-) mice

After characterizing the beneficial effects in hepatic metabolic dysregulation in fructose-exposed *aldob/ampd2*(-/-) mice, we then conducted an additional study to determine whether deleting AMPD2 increased the tolerance and preference for fructose of in *aldob*(-/-) mice. To this end, we measured intake of fructose and sugar-containing solutions. These solutions included fructose, sucrose and fructose: glucose combination and contained the same concentration of fructose (2.5% w/v). In single bottle (no choice) intake study, *aldob*(-/-) mice consumed the least amount of any solution tested. Specifically, in 24 h drinking tests and on average, *aldob*(-/-) mice consumed $0.11 \pm 0.10$ ml of fructose, $0.21 \pm 0.11$ ml of sucrose and $0.08 \pm 0.02$ ml of HFCS. In contrast, wild type mice demonstrated high appetite for fructose-containing sugars ($5.08 \pm 0.90$ ml of fructose, $7.43 \pm 0.99$ ml of sucrose and $12.62 \pm 1.30$ ml of HFCS). Of interest, deletion of AMPD2 in *aldob*(-/-) mice significantly increased the consumption of sugary drinks ($2.31 \pm 0.46$ ml of fructose, $3.55 \pm 0.71$ ml of sucrose and $2.65 \pm 0.70$ ml of HFCS, $P < 0.01$ vs *aldob*(-/-)) (Fig. 3a). Of note, *aldob/ampd2*(-/-) mice did not consume as much sugar as will type mice suggesting that correcting metabolic dysregulation is not sufficient to stimulate sugar intake in mice deficient for AMPD2 and that other side-mechanisms are taking place to control sugar intake. Similarly, when mice were switched from a sucrose-free diet to regular chow containing 1% fructose, *aldob/ampd2*(-/-) mice consumed significantly more chow than control *aldob*(-/-) animals ($1.06 \pm 0.33$ g/day in *aldob*(-/-) vs $2.03 \pm 0.32$ g/day in *aldob/ampd2*(-/-) mice, $P < 0.01$, Fig. 3b). These observations suggest that *aldob/ampd2*(-/-) mice have improved tolerance to fructose in voluntary drinking. Consistently, and unlike wild type or *aldob/ampd2*(-/-) mice, *aldob*(-/-) animals demonstrate a marked failure to thrive when maintained on a 1% fructose diet for 5 consecutive weeks with a greater than 7% reduction in total body weight (Fig. 3c, d). Failure to thrive and growth retardation are common consequences in subjects with HFI, particularly in babies after weaning and first exposed to this sugar[35,36]. The failure to thrive can be the consequence of a low energy state, reduced caloric intake as suggested in Figs. 1f and 3b, but also to the metabolic consequences associated with Fanconi syndrome, a defect common in subjects with HFI and the result of renal tubular acidosis and urinary glucose turnover[14,24,37–40]. We and others have shown that uric acid, as a by-product of fructose metabolism is an important contributor to the Fanconi syndrome[41–44] and hyperuricemia is common in HFI[22,32]. Consistently, plasma uric acid levels were significantly higher in 1% fructose fed *aldob*(-/-) mice compared to control and *aldob/ampd2*(-/-) (Fig. 3e). Furthermore, fractional excretion of uric acid, and phosphate as well as

**Fig. 1 | Activation of AMPD2 and hepatic metabolic dysregulation in *aldob*(-/-) mice. a** Proposed schematic of AMPD2 activation in HFI. In wild type (*aldob*(+/+)) mice, fructose is metabolized to fructose-1-phosphate (F1P) by fructokinase (KHK) and to glyceraldehyde and dihydroxyacetone phosphate (DHAP) by aldolase B. However, in *aldob*(-/-) mice, the accumulation of F1P leads to intracellular phosphate depletion and the activation of AMPD2. AMPD2 deaminates AMP to inosine monophosphate (IMP) in the first step of the purine degradation pathway which in humans results in uric acid formation. **b** Representative western blot from liver extracts for AMPD2, aldolase B and Actin in wild type and *aldob*(-/-) mice. **c** Intracellular phosphate levels in wild type (*aldob*(+/+)) and *aldob*(-/-) mice fed sucrose-free (W) or chow containing 1% (w/w) fructose (F) for 2 weeks. **d** Hepatic AMPD activity in the same mouse groups as in **c**. **e–g** Intrahepatic ATP, ADP and AMP levels and calculated energy charge and AMP/ATP ratio in the same mouse groups as in **c**. **h** Representative western blot from liver extracts for activated (pAMPK), total AMPK and Actin control in the same groups as in (**c**). **i** Top, representative western blot from liver extracts for inhibited (pGS), total GS and Actin control in 1% fructose-fed wild type and *aldob*(-/-) mice. Bottom, representative liver PAS image from 1% fructose-fed wild type and *aldob*(-/-) mice. Blue arrows denote macrosteatotic areas. **j** Representative western blot from liver extracts for inhibited (pACC), total ACC and Actin control in 1% fructose-fed wild type and *aldob*(-/-) mice. **k** Representative liver H&E image from 1% fructose-fed wild type and *aldob*(-/-) mice. **l** Liver triglycerides in wild type (*aldob*(+/+)) and *aldob*(-/-) mice fed sucrose-free or chow containing 1% (w/w) fructose for 2 weeks. Blue arrows denote macrosteatotic areas. Red arrows indicate areas with ductal reaction. Size Bar: 20 μM. PT Portal triad, CV Central vein. The data in (**b–g**) were presented as the means ± SEM and analyzed by One Way ANOVA with Tukey post hoc analysis. *n* = 6 mice per group.

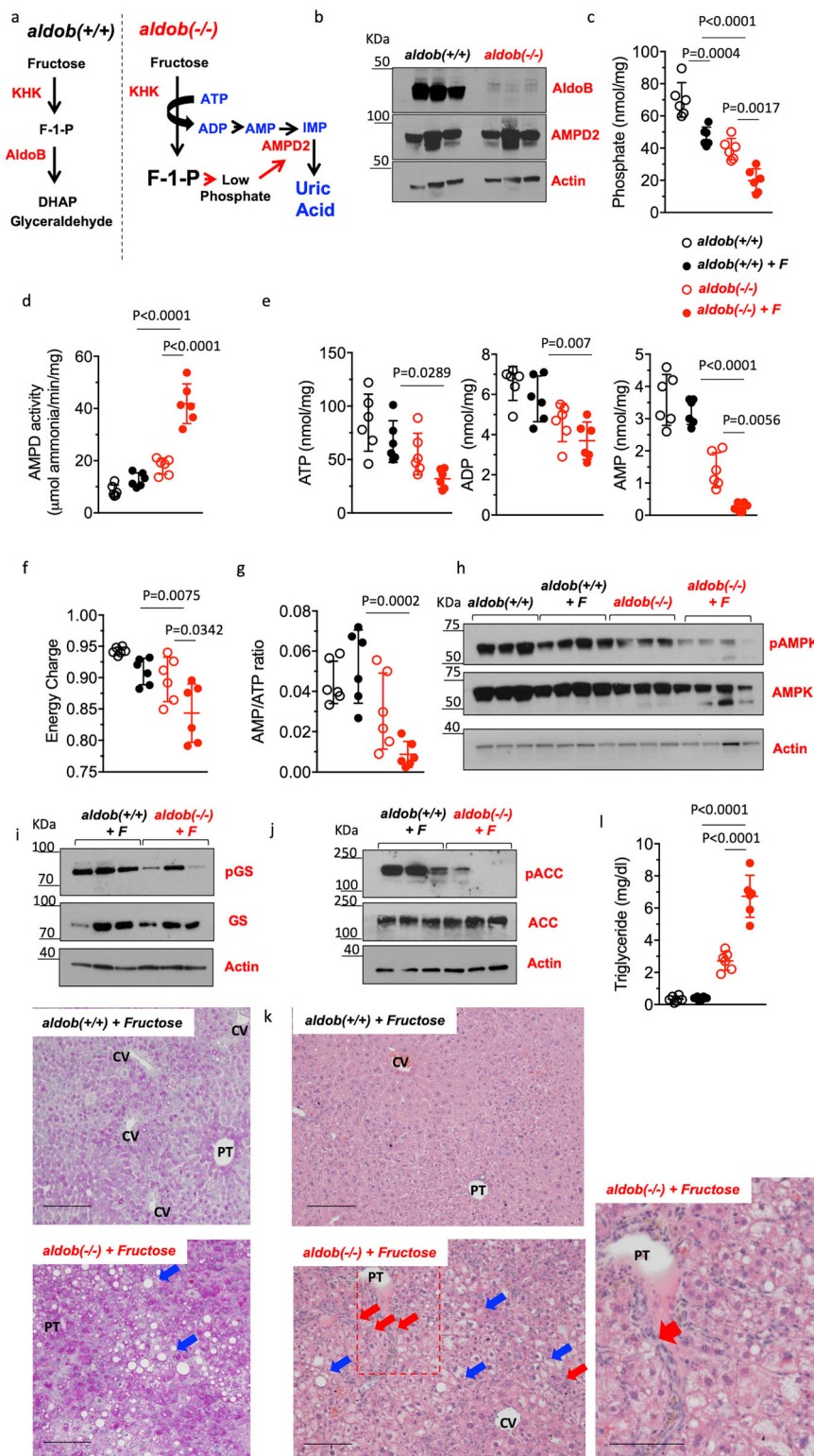

glucosuria with no major histological pathology in the proximal tubule, all markers of Fanconi syndrome and renal dysfunction were significantly elevated in *aldob*(-/-) mice compared to control and *aldob/ampd2*(-/-) mice (Fig. 3f–i).

Finally, to determine whether uric acid could contribute to the retardation in growth gain induced by fructose in *aldob*(-/-), we then further increased plasma uric acid with oxonic acid (2.5% w/w) a selective uricase inhibitor that prevents the degradation of uric acid to allantoin[45,46], (Fig. 4a). As shown in Fig. 4b, oxonic acid efficiently elevated plasma uric acid levels in

fructose-exposed *aldob*(-/-) mice (2.26-fold, *P* < 0.01) which caused a greater failure to thrive in fructose-fed mice. While exposure to fructose resulted in 6.1 ± 1.4% in body weight loss, the combination of fructose and oxonic acid decreased body weight by 12.4 ± 2.8% (*P < 0.05*, Fig. 4c). These changes in uric acid were associated with pathological findings including interstitial inflammation, presence of pigmented macrophages and mesangial expansion in glomeruli (Fig. 4d) and greater nutrient waste into the urine as denoted by higher fractional excretion of uric acid and phosphate (Fig. 4e, f) and glucosuria (Fig. 4g).

**Fig. 2 | Blockade of AMPD2 ameliorates metabolic dysregulation in** *aldob*(-/-) **mice. a** Representative western blot from liver extracts for AMPD2, aldolase B and Actin control in wild type (*aldob*(+/+)), *aldob*(-/-) control, AMPD2 heterozygous and AMPD2 homozygous mice. Hepatic IMP (**b**), AMP (**c**), energy charge (**d**) and AMP/ATP ratio (**e**) in the same mouse groups as in (**a**). **f** Representative western blot from liver extracts for activated (pAMPK), total AMPK and Actin control in the same groups as in **a**. Representative liver PAS image (**g**) in fructose-fed and glycogen content (**h**) in the same groups as in (**g**). Blue arrows denote macrosteatotic areas. Liver triglyceride content (**i**) in the same groups as in (**g**) and representative H&E image (**j**) in fructose-fed *aldob*(-/-) and *aldob/ampd2*(-/-) mice. Blue arrows denote macrosteatotic areas. Red arrows indicate inflammation. Insert shows presence of pigmented macrophages. **k** Representative picro-sirius red under brightfield (left) and polarized light (right) images in fructose-fed *aldob*(-/-) and *aldob/ampd2*(-/-) mice. Liver hydroxyproline (**l**), liver injury score (**m**) and liver transaminases (**n**) in the same groups as in (**g**). Size Bar: 20 μM. PT Portal Triad. CV Central Vein. The data in (**b**–**e**), **h**–**i** and **l**–**n** were presented as the means ± SEM and analyzed by One Way ANOVA with Tukey post hoc analysis. *n* = 6 mice per group.

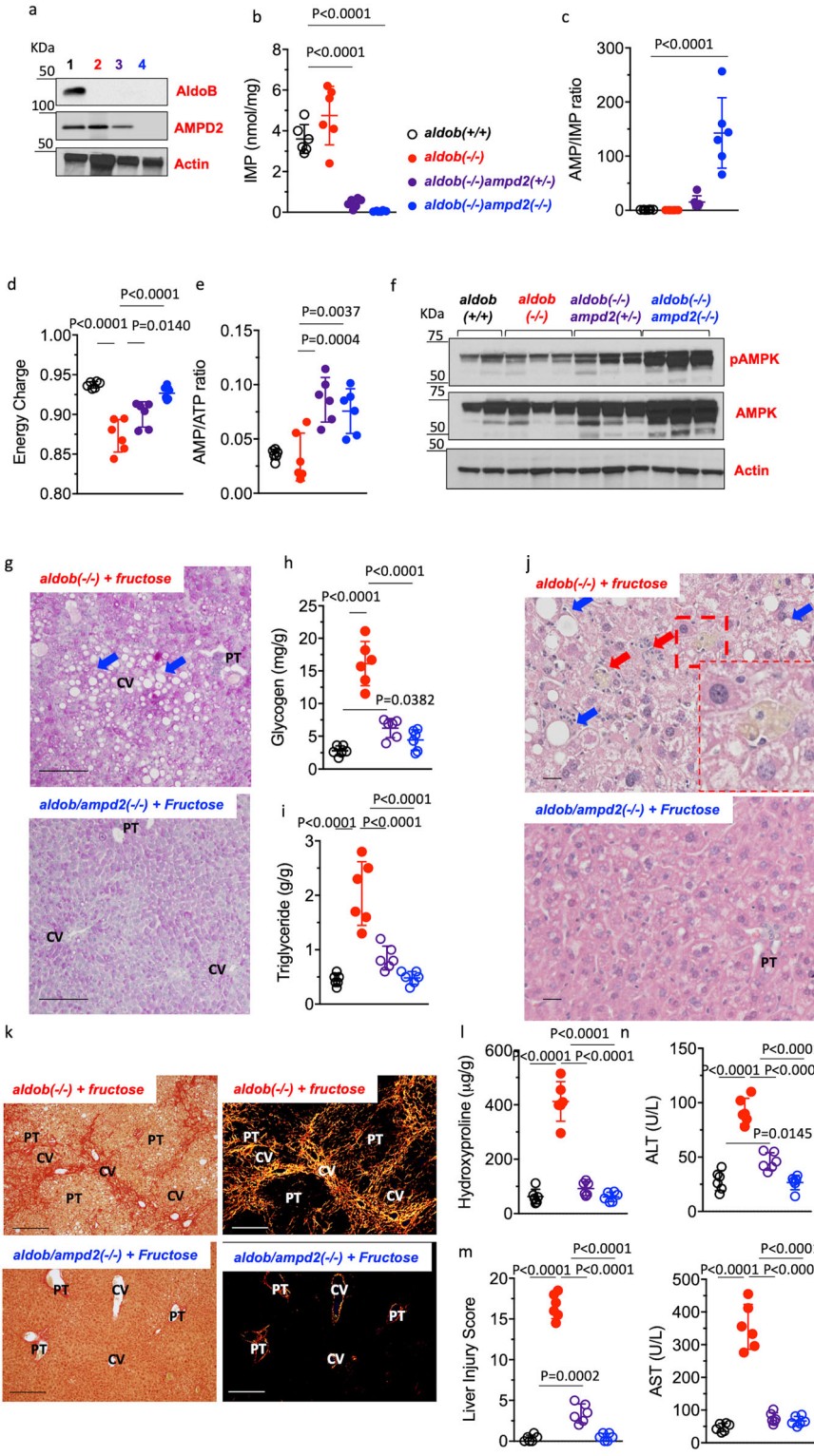

## Specific hepatocyte deletion of AMPD2 prevents liver injury in fructose-fed aldob(-/-) mice

AMPD2 is expressed in several tissues important for nutrient sensing and metabolism. To specifically determine the liver metabolic malfunction associated with fructose in *aldob*(-/-) mice, we conducted a subsequent study in which AMPD2 expression was deleted in hepatocytes as we have previously described[26]. Specific AMPD2 deletion in the liver of *aldob*(-/-) mice (*aldob/ampd2*[Fl/Fl]*xAlbCre*(-/-)) is shown in Fig. 5a. After 5 weeks exposure of mice to chow containing 1% (w/w) fructose, and consistent with the data

obtained in whole body AMPD2 deficient mice in Figs. 2 and 3, deletion of AMPD2 in hepatocytes resulted in a marked improvement in liver dysfunction as denoted histologically (Fig. 5b–d) and improved markers of liver dysfunction including plasma ALT (Fig. 5e) and glycogen content (Fig. 5f). Furthermore, liver fibrosis in response to fructose was markedly reduced in *aldob/ampd2*[Fl/Fl]*xAlbCre*(-/-) as demonstrated by both histological staining of picrosirius red (PSR) (Fig. 5g) and liver hydroxyproline levels (Fig. 5h).

In Table 1 we have compiled and analyzed statistical significance of all liver and plasma parameters measured in the strains in sucrose-free chow or

**Fig. 3 | Blockade of AMPD2 improves the tolerance to fructose of *aldob*(-/-) mice. a** Average daily intake of 5% (w/v) fructose, sucrose and high fructose corn syrup (HFCS) solutions in wild type (*aldob*(+/+)), *aldob*(-/-) and *aldob/ampd2*(-/-) mice. **b** Average daily intake sucrose-free (Suc free) or 1% fructose-containing chow in wild type (*aldob*(+/+)), *aldob*(-/-) and *aldob/ampd2*(-/-) mice. Body weight (**c**) and body weight change (**d**) in sucrose-free maintained diet and after switching to a 1% fructose diet in wild type (*aldob*(+/+)), *aldob*(-/-) and *aldob/ampd2*(-/-) mice. **e** Plasma uric acid levels in wild type (*aldob*(+/+)), *aldob*(-/-) and *aldob/ampd2*(-/-) mice fed 1% fructose chow for 5 weeks. Fractional excretion of uric acid (**f**), phosphate (**g**) and urinary glucose excretion (**h**) in the same groups as in (**e**). **i** Representative kidney PAS image in sucrose-free and 5-week 1% fructose-fed *aldob*(-/-) mice. Size Bar: 20 μM. Blue arrows denote tubules with minimal cast. The data in (**a**–**h**) were presented as the means ± SEM and analyzed by One Way ANOVA with Tukey post hoc analysis. *n* = 6 mice per group Illustration created with royalty-free images obtained from pixabay (2018) (www.pixabay.com).

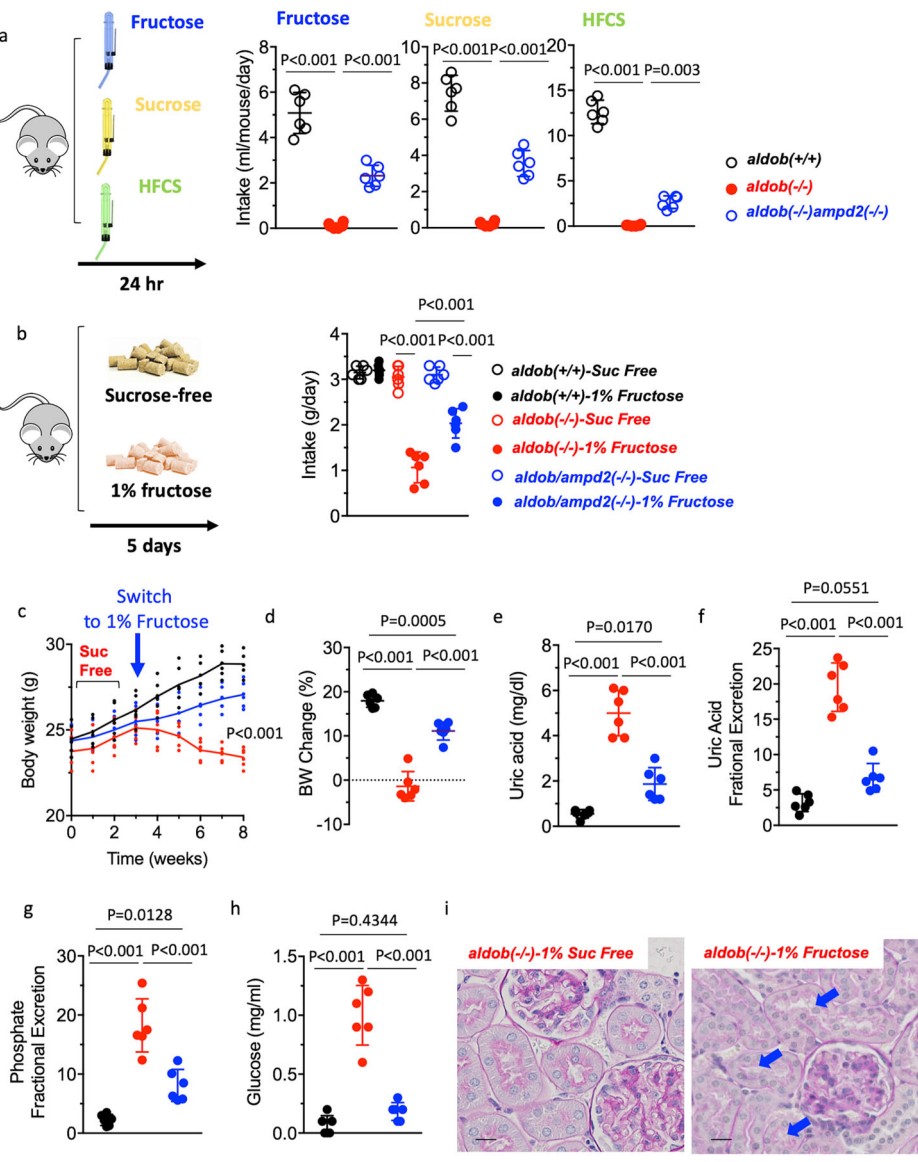

fed 1% (w/w/) fructose for 5 weeks. Strains include wild type (*aldob*(+/+)), *aldob*(-/-), *aldob/ampd2*(-/-) and *aldob/ampd2*$^{Fl/Fl}$*xAlbCre*(-/-).

## Acute hypoglycemia in HFI is a F1P-dependent, AMPD2-independent effect

Hypoglycemia is a serious complication of HFI detected after acute ingestion of high amounts of fructose. We previously demonstrated that acute hypoglycemia in *aldob*(-/-) mice is mediated by the translocation of glucokinase (GCK) from the nucleus to the cytosol triggering glycolysis and glucose uptake[24]. Fructose-1-phosphate (F1P) levels after an acute fructose challenge (1.5 mg/g BW) revealed high but no significant differences between *aldob*(-/-) and *aldob/ampd2*(-/-) mice (Fig. 6a). Thus, and consistent with a F1P specific effect, plasma glucose levels dramatically dropped in *aldob*(-/-) mice independently of AMPD2 expression (Fig. 6b) in association with an increased translocation of GCK to the cytosol (Fig. 6c).

In summary, our data support a mechanism driving HFI similar to the one depicted in Fig. 6d in which the non-functional mutation of aldob promotes two major effects, the accumulation of F1P (phosphate trap) and the activation of AMPD2 and the purine degradation pathway. Both events are connected as the build-up and accumulation of F1P activates AMPD2 and contributes to create a low energy state due to the sequestration of phosphate. However, the consequence and relevance of each mechanism is

different. In our proposed mechanism, hypoglycemia is the only F1P-dependent and AMPD2-independent condition in HFI and relies on the rapid ingestion of high amounts of fructose. In contrast, the activation of AMPD2 by much lower amounts of fructose, functionally contributes to a much greater pathology characterized by metabolic dysregulation and energy imbalance with low mobilization of energy stores (glycogen and fat), inflammation, Fanconi syndrome, nutrient loss (glucose, amino acids and phosphate) in the urine and failure to thrive/growth retardation. Presumably, the loss of phosphate in the urine results in hypophosphatemia[47] which further impairs the ability of the liver to replenish phosphate from the circulation.

## Discussion

Hereditary fructose intolerance (HFI) is an orphan genetic disease with no treatment to date caused by a loss-of-function mutation in *aldob*. Aldob is a key enzyme for the metabolism and processing of fructose. The only option in management is avoidance of dietary intake of fructose, but this is exceptionally difficult given that sugar or HFCS are present in 75 percent of processed foods in the United States[48]. Because of this, subjects with HFI have "learned" to avoid foods containing fructose and fructose-containing-sugars. However, even the exclusion of foods containing sugar or HFCS from the diet do not totally eliminate chronic fructose intoxication, and the

**Fig. 4 | Raising uric acid exacerbates growth retardation in fructose-exposed aldob(-/-) mice.**
**a** Schematic of the effect of targeting uricase (UOx) by oxonic acid (OxAc) in mice. **b** Plasma uric acid levels in *aldob*(-/-) (KW) mice control (red) or on oxonic acid (purple) fed sucrose-free or 1% fructose diets for 5 weeks. **c** Body weight in the same groups as in (**b**). **d** Representative kidney PAS image in 1% fructose alone or in combination with oxonic acid fed *aldob*(-/-) mice. Yellow arrows point to areas of interstitial inflammation and pigmented macrophages. Red circle indicate glomerular mesangial expansion. Size Bar: 20 μM. Fractional excretion of uric acid (**e**), phosphate (**f**) and urinary glucose excretion (**g**) in the same groups as in (**b**). The data in (**b**, **c** and **e**–**g**) were presented as the means ± SEM and analyzed by One Way ANOVA with Tukey post hoc analysis. *n* = 6 mice per group.

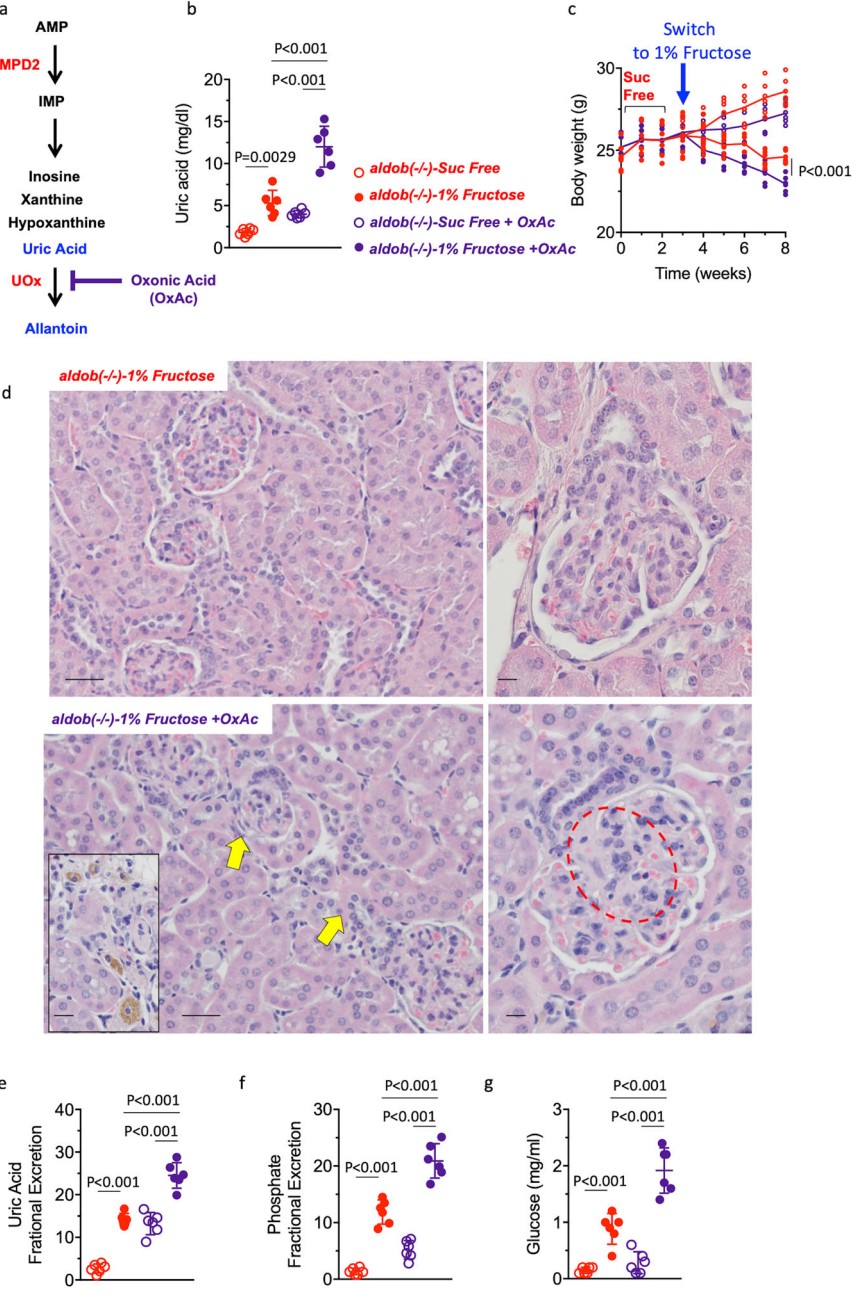

general ill health described in people with aldolase B deficiency[49]. This observation reflects the high sensitivity of people with HFI to fructose to whom even traces of fructose are sufficient to trigger a very dramatic pathological phenotype. Here, we provide evidence that AMPD2 and the purine degradation pathway is a critical pathogenic pathway in HFI. We hypothesize that the activation of the AMPD2 pathway by fructose is responsible for the majority of the deleterious consequences induced by this sugar in HFI and therefore, we provide evidence, at multiple levels, of a pathway that can be targeted to treat HFI.

Two major discoveries are drawn from our study, First, that the activation of AMPD2 is a key driver of many of the manifestations of HFI and induced by relatively low amounts of fructose. In our study, we have used a 1% fructose chow (w/w) diet which is very similar to the amount in most rodent diets. In this regard, mice are naturally resistant to the effects of fructose and require much higher doses, greater than 35% of the total caloric intake[50,51], to induce significant metabolic dysfunction, especially when provided in the chow. This indicates that the deleterious effects of fructose in

HFI can be caused by very low amounts and further emphasizes the importance of avoiding fructose in foods by subjects with this condition, which is very challenging as fructose is added to many foods to increase its palatability and stimulate intake.

The activation of the AMPD2-driven side-chain in HFI provides a major insight on how fructose promotes hepatic fat accumulation. It is well accepted that fructose triggers de novo lipogenesis[52–54]. However, mice and humans with HFI are not able to fully convert fructose to fat and yet fatty liver and non-alcoholic steatohepatitis are common findings in HFI. Similarly, isotopic tracer studies indicate that only a small fraction of fructose (<1%) is directly converted into fat[55]. Our findings suggest that it is the metabolic dysregulation initiated by AMPD2 and not the full processing of fructose into fat that drives its lipogenic actions. AMPD2 and the purine degradation pathway, by modulating AMPK activity or the energy state would regulate expression and activation of lipogenic enzymes like ACC and thus would control lipogenic rates. This is consistent with previous reports demonstrating a key role for uric acid in promoting de novo

**Fig. 5 | Liver-specific deletion of AMPD2 prevents liver injury in fructose-fed *aldob*(-/-) mice.**
**a** Representative western blot from liver extracts for AMPD2, aldolase B and Actin control in wild type (*aldob*(+/+)) and *aldob/ampd2^FI/FL^xAlbCre*(-/-) mice in kidney (K), liver (L), duodenum (Duo) and jejunal (Jej) extracts. **b** Representative liver H&E image in sucrose-free and fructose-fed *aldob*(-/-) and *aldob/ampd2^FI/FL^xAlbCre*(-/-) mice. Blue arrows denote macrosteatotic areas. Red arrows indicate inflammation. **c** Liver triglyceride content. **d** Liver injury score. **e** Plasma ALT. **f** Liver glycogen **g** Representative picro-sirius red images under brightfield (left) and polarized light (right) and h) Liver hydroxyproline levels in sucrose-free and fructose-fed *aldob*(-/-) and *aldob/ampd2^FI/FL^xAlbCre*(-/-) mice. Size Bar: 20 μM. PT Portal Triad. CV Central Vein. The data in (**c**–**f**) and **h**) were presented as the means ± SEM and analyzed by One Way ANOVA with Tukey post hoc analysis. *n* = 6 mice per group.

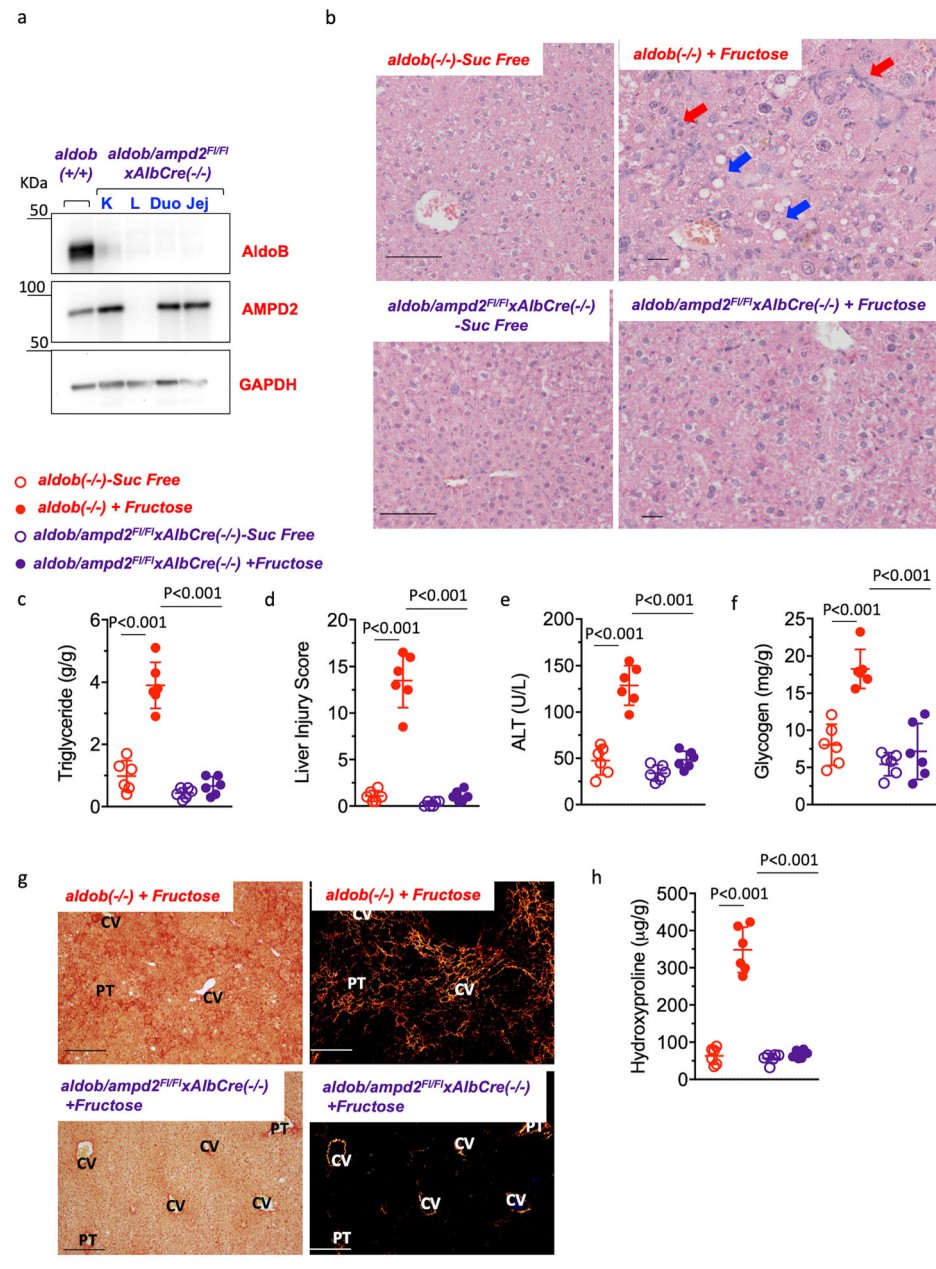

lipogenesis[56–58] and the recent report from Zhao et al.[59] demonstrating a two-pronged mechanism regulating hepatic lipogenesis by fructose, in which hepatocytes promote gene expression and activity of lipogenic enzymes (including ACC and fatty acid synthase), while microbial fructose-derived acetate is the actual lipogenic substrate. It is important to note that other enzymes of the purine degradation pathway besides AMPD2 may be implicated in the pathogenesis induced by fructose in *aldob*(-/-) mice. For example, adenosine, produced from AMP under conditions of low energy charge like hypoxia or inflammation, promotes cytoprotective actions in response to the low energy state in many tissues including the liver[60]. Our data suggests that the energy charge in fructose-exposed *aldob*(-/-) is particularly low and thus adenosine may be important to protect cells from energy depletion. Therefore, excessive elimination of adenosine via adenosine deaminase may exert important deleterious consequences in HFI. Further, adenosine metabolism via adenosine deaminase increases the production of metabolites in the purine degradation pathway like inosine which in turn thus contributing to uric acid production and hyperuricemia.

The second major finding in our study is the importance of phosphate in the regulation of liver metabolism. Our results indicate the presence of a

"phosphate trap" induced by fructose in HFI which ultimately triggers the activation of AMPD2. In these settings, low phosphate would be the result of both its sequestration as F1P and increased turnover in the circulation as a consequence of renal tubular acidification and high urinary fractional excretion of phosphate. In this regard, recent epidemiological evidence demonstrates an inverse correlation between phosphate levels, BMI and percentage of fat[61], low phosphorus intake is involved in weight gain and metabolic syndrome[62,63] and subjects with genetic hypophosphatemia have increased prevalence of obesity and higher BMI[64]. This would suggest that therapies aimed to return to homeostatic levels of phosphate may be clinically relevant in HFI. These therapeutic approaches would include both dietary phosphorus supplementation and the use of insulin-sensitizing agents to promote phosphate uptake in liver and inhibit AMPD activity as we have previously shown that work in the skeletal muscle[25,65]. It is important to note that besides the activation of AMPD2 and the purine degradation pathway, it is likely that other alternative or complementary pathways are been modified that explain the overall metabolic phenotype observed in fructose-exposed *aldob*(-/-) mice. In this regard, tracer and metabolomics analysis would be an essential tool to determine

**Table 1 | Metabolic parameters in mouse strains in sucrose-free or 1% fructose diet**

| | aldoB(+/+) | | aldoB(-/-) | | aldoB /AMPD2(-/-) | | aldoB/AMPD2$^{Fl/Fl}$xAlbCre | |
|---|---|---|---|---|---|---|---|---|
| | Sucrose-free (n = 6) | Fructose (n = 6) | Sucrose-Free (n = 6) | Fructose (n = 6) | Sucrose-Free (n = 6) | Fructose (n = 6) | Sucrose-free (n = 6) | Fructose (n = 6) |
| *Body weight and composition* | | | | | | | | |
| Body weight; 0 W (g) | 25.9 ± 1.3 | 26.1 ± 1.2 | 25.6 ± 0.4 | 25.2 ± 1.0 | 25.1 ± 0.5 | 25.5 ± 1.1 | 26.4 ± 0.4 | 25.7 ± 0.6 |
| Body weight; 5 W (g) | 26.9 ± 1.4 | 28.8 ± 1.0 | 26.1 ± 0.6 | 23.2 ± 0.7 | 26.9 ± 0.4 | 27.7 ± 1.1 | 28.2 ± 0.6 | 28.1 ± 0.4 |
| ΔBody weight; 5 W (g) | 1.0 ± 0.5 | 2.7 ± 0.4 | 0.5 ± 0.6 | -2.0 ± 0.5 | 1.8 ± 0.4 | 2.2 ± 1.1 | 1.8 ± 0.6 | 2.4 ± 0.6 |
| Average Food Intake (g/mouse/day) | 3.1 ± 0.3 | 3.1 ± 0.2 | 2.8 ± 0.2 | 2.3 ± 0.3 | 2.9 ± 0.2 | 2.7 ± 0.3 | 3.0 ± 0.2 | 3.1 ± 0.2 |
| Liver weight; 5 W (g) | 1.46 ± 0.12 | 1.52 ± 0.06 | 1.41 ± 0.13 | 1.47 ± 0.05 | 1.53 ± 0.09 | 1.52 ± 0.06 | 1.53 ± 0.6 | 1.56 ± 0.4 |
| Liver/Body weight ratio | 0.054 ± 0.02 | 0.052 ± 0.01 | 0.054 ± 0.02 | 0.063 ± 0.02 | 0.056 ± 0.02 | 0.054 ± 0.02 | 0.054 ± 0.02 | 0.055 ± 0.02 |
| Kidney weight; 5 W (g) | 0.40 ± 0.05 | 0.42 ± 0.04 | 0.47 ± 0.03 | 0.52 ± 0.01 | 0.43 ± 0.05 | 0.42 ± 0.04 | 0.42 ± 0.04 | 0.44 ± 0.04 |
| Kidney/Body weight ratio | 0.014 ± 0.01 | 0.014 ± 0.01 | 0.018 ± 0.01 | 0.022 ± 0.01 | 0.015 ± 0.01 | 0.015 ± 0.01 | 0.015 ± 0.01 | 0.015 ± 0.01 |
| *Biochemical blood analysis* | | | | | | | | |
| AST (IU/L) | 47.2 ± 5.2 | 46.8 ± 3.7 | 64.6 ± 11.6 | 354.2 ± 27.8 | 52.3 ± 5.0 | 58.1 ± 6.2 | 36.9.0 ± 11.1 | 48.2 ± 7.8 |
| ALT (IU/L) | 47.5 ± 6.2 | 42.6 ± 5.7 | 66.7 ± 6.6 | 128.7 ± 8.7 | 41.4 ± 7.6 | 38.4 ± 8.4 | 48.3 ± 3.8 | 46.6 ± 6.8 |
| Serum triglycerides (mg/dl) | 66.7 ± 12.6 | 75.8 ± 7.2 | 52.6 ± 7.2 | 32.5 ± 5.2 | 61.6 ± 7.4 | 65.2 ± 5.7 | 68.0 ± 6.3 | 59.7 ± 7.2 |
| Serum uric acid (mg/dl) | 1.56 ± 0.1 | 1.65 ± 0.1 | 1.69 ± 0.1 | 2.33 ± 0.1 | 1.19 ± 0.1 | 1.16 ± 0.1 | 1.23 ± 0.1 | 1.31 ± 0.2 |
| Fasting serum glucose (mg/dl) | 116 ± 8.8 | 122 ± 8.5 | 106 ± 4.5 | 92 ± 4.8 | 114 ± 7.5 | 112 ± 6.9 | 116 ± 6.1 | 113 ± 5.1 |
| Fasting insulin (ng/ml) | 0.55 ± 0.20 | 0.42 ± 0.2 | 0.36 ± 0.2 | 0.32 ± 0.1 | 0.46 ± 0.1 | 0.41 ± 0.1 | 0.45 ± 0.1 | 0.48 ± 0.1 |
| *Biochemical Urine Analysis* | | | | | | | | |
| Glucose (mg/ml) | 0.06 ± 0.03 | 0.03 ± 0.03 | 0.11 ± 0.04 | 1.03 ± 0.13 | 0.02 ± 0.01 | 0.03 ± 0.01 | 0.04 ± 0.01 | 0.04 ± 0.01 |
| Fractional excretion of phosphate | 2.22 ± 0.32 | 2.34 ± 0.21 | 3.12 ± 0.32 | 18.25 ± 1.83 | 1.18 ± 0.27 | 8.1 ± 1.1 | 2.51 ± 0.16 | 5.34 ± 1.30 |
| Fractional excretion of urate | 3.2 ± 0.51 | 3.4 ± 0.61 | 4.2 ± 0.66 | 19.6 ± 1.39 | 3.6 ± 0.23 | 6.7 ± 0.83 | 2.9 ± 0.22 | 4.7 ± 0.67 |
| *Liver analysis* | | | | | | | | |
| Glycogen (mg/g) | 2.36 ± 0.62 | 2.78 ± 0.72 | 6.23 ± 0.91 | 16.13 ± 3.38 | 2.52 ± 0.44 | 3.49 ± 1.57 | 2.16 ± 0.16 | 2.51 ± 0/33 |
| Triglycerides (g/g) | 0.32 ± 0.15 | 0.45 ± 0.05 | 1.21 ± 0.34 | 2.11 ± 0.46 | 0.54 ± 0.11 | 0.85 ± 0.21 | 0.43 ± 0.16 | 0.66 ± 0.29 |
| Energy charge | 1.11 ± 0.03 | 1.14 ± 0.02 | 0.70 ± 0.02 | 0.50 ± 0.03 | 1.01 ± 0.04 | 1.06 ± 0.03 | 1.02 ± 0.02 | 1.24 ± 0.03 |
| Hydroxyproline (μg/g) | 47.3 ± 7.62 | 53.6 ± 7.62 | 63.1 ± 9.14 | 348.2 ± 25.2 | 51.5 ± 4.18 | 58.2 ± 6.33 | 56.8 ± 5.28 | 67.1 ± 4.40 |

**Fig. 6 | AMPD2 deletion do not protect mice from fructose-dependent acute hypoglycemia.**
**a** Intrahepatic fructose-1-phosphate (F1P) levels in wild type (*aldob*(+/+)), *aldob*(-/-) and *aldob/ampd2*(-/-) at baseline or 120' after receiving 1.5 mg/g fructose by oral gavage. **b** Continuous plasma glucose (% change from baseline) levels in wild type (*aldob*(+/+)), *aldob*(-/-) and *aldob/ampd2*(-/-) mice after receiving 1.5 mg/g fructose by oral gavage. **c** Hepatic nuclear and cytosolic glucokinase (GCK) and nuclear marker control (CREB) expression in wild type (*aldob*(+/+)), *aldob*(-/-) and *aldob/ampd2*(-/-) mice after receiving 1.5 mg/g fructose by oral gavage. **d** Proposed schematic on the differential effects of fructose metabolism in *aldob*(-/-) mice. Fructose metabolism in aldob(-/-) mice leads to both accumulation of F1P and the activation of AMPD2 via phosphate depletion. F1P causes acute hypoglycemia in response to high fructose exposure via activation of GCK and glycolysis. On the other hand, AMPD2 activation at low fructose concentrations triggers hepatic metabolic dysregulation characterized by low energy charge, defective glycogen and lipid metabolism and the production and accumulation of uric acid. In the kidney, renal tubular acidosis upon fructose exposure is manifested by Fanconi syndrome and inefficient tubular reabsorption leading to nutrient waste and failure to thrive. Illustration created with royalty-free images obtained from pixabay (2018) (www.pixabay.com). The data in **a**, **b** were presented as the means ± SEM and analyzed by One Way ANOVA with Tukey post hoc analysis. For a, \*\**P* < 0.01 versus respective vehicle-gavage control. For B), \*\**P* < 0.01 versus fructose-gavage wild type (1.5 mg/g). *n* = 6 mice per group.

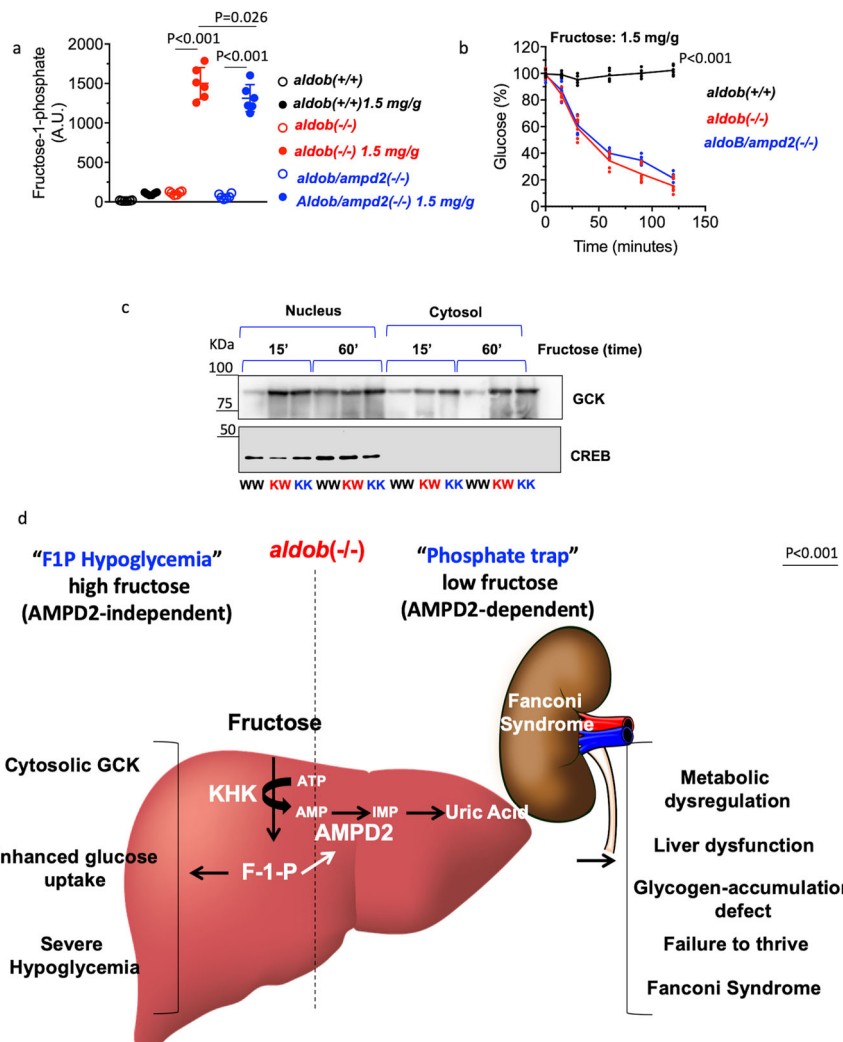

malfunctioning metabolic pathways and would be particularly informative in the HFI process to slow the development and progression of the disease.

Together, our data from *aldob*(-/-) mice may also be relevant to the general population with regards to features of metabolic syndrome. Based upon the results here, we propose that conditions like fatty liver, fat accumulation, dyslipidemia or renal dysfunction could be an HFI-like response to excessive and chronic exposure to fructose.

## Methods

Study approval: All animal experiments were conducted with adherence to the National Institutes of Health *Guide for the Care and Use of Laboratory Animals*. The animal protocol (#1253, PI. Lanaspa) was approved by the Institutional Animal Care and Use Committee of the University of Colorado (Aurora, CO, USA). We have complied with all relevant ethical regulations for animal use.

Animal experiments: Experiments were carried in young adult (8 weeks old) male and female mice. *Aldob*(-/-) mice in a C57BL/6 J genetic background were generated by and obtained from Dr Dean Tolan at Boston University[23]. AMPD2 knockout, *ampd2*(-/-) mice (B6.129-Ampd2[tm1Tm]/J) were originally obtained from the Jackson Laboratory and a colony is maintained at the University of Colorado. *aldob/ampd2*(-/-) and wild type controls were obtained from *aldob/am*pd2(+/-) breeding pairs fed a sucrose-free chow (Bioserv, catalog F6700) to ensure no fructose pre-conditioning was induced in the offspring. Carbohydrate content in sucrose-free chow consists of 50% (w/w) dextrose and 12.9% (w/w) corn

starch. Conditional ampd2 knockout mice were developed by Ingenious Targeting Laboratory (Ronkonkoma, NY) using the Cre-Lox system (ampd2[Fl/Fl]). To this end, exons 2–9 were flanked by LoxP sites for cre-recombinase specific deletion. Tissue specific Cre-expressing mice were purchased from Jackson laboratories, Cre-Albumin (018961). *AldoB/ampd2[Fl/Fl]xAlbCre*(-/-) were produced as in Andres-Hernando et al.[26] For experiments, mice (female and male) were maintained in temperature- and humidity-controlled specific pathogen-free conditions on a 14 h dark/10 h light cycle and depending on the study, allowed ad libitum access to 1% fructose-containing regular chow (Harlan Teklad, #2920X), 2.5% oxonic acid chow (w/w) or fructose-free chow. Food and sugary water (fructose, sucrose or high fructose corn-syrup, 5% (w/v)) were measured daily. An a priori power analysis was not performed for determining sample size, but instead prior experience with the acute and chronic fructose-exposure of *aldob*(-/-) mouse model[20,24] was used to establish a sample size sufficient to delineate therapeutic efficacy. Mice were randomly assigned to the dietary regimens using the Random function in Excel and all samples/slides were analyzed in a blinded fashion and then decoded, grouped and the results analyzed. Very importantly, no mice were excluded from the study due to health concerns or any other criterion.

Histopathology: Formalin-fixed paraffin-embedded liver and kidney sections were stained with hematoxyline and eosine (H&E) for liver injury scoring and lipid assessment. Sections were stained with periodic acid–Schiff (PAS) for kidney injury scoring and glycogen assessment, and with Picro-Sirius Red (PSR) for fibrosis assessment. For liver injury scoring,

the entire cross section of liver was analyzed from each mouse using a variation of the Brunt system[66] modified for mouse liver[24]. For kidney injury scoring, cross sections of both kidneys were examined and scored as recently described[67]. For assessment of fibrosis, 10 high quality images were made in a tiling fashion across the liver and imported into the next generation Image J program named SlideBook (Intelligent Imaging Innovations, Denver Colorado) and here positive pixels were assessed as a percent of the total pixels[24]. Images were captured on an Olympus BX51 microscope equipped with a 4 megapixel Macrofire digital camera (Optronics) using the PictureFrame Application 2.3 (Optronics). Composite images were assembled with the use of Adobe Photoshop. All images in each composite were handled identically.

## Western blotting

Protein content was determined by the bicinchoninic acid (BCA) protein assay (Pierce). Total protein (50 μg) was separated by SDS-PAGE (10% w/v) and transferred to PVDF membranes (BioRad). Membranes were first blocked for 1 h at 25 °C in 4% (w/v) instant milk dissolved in 0.1% Tween-20 Tris-buffered saline (TTBS), and incubated with the following primary rabbit or mouse-raised antibodies (1:1,000 dilution in TTBS, with catalog numbers indicated): aldolase B (H00000229-A01), GCK (H000026465-B02P), ACC (3676), p-ACC (Ser79, 11818), AMPK (2603), pAMPK (Thr172, 2535), GS (3886), pGS (Ser641 3981) and Actin (4970) (all from Cell Signaling); and AMPD2 (Abnova 271). Membranes were visualized using an anti–rabbit (7074) or anti–mouse IgG (7076) horseradish peroxidase–conjugated secondary antibody (1:2,000, Cell Signaling) using the HRP Immunstar detection kit (Bio-Rad). Chemiluminescence was recorded with an Image Station 440CF and the results were analyzed with 1D Image Software (Kodak Digital Science). Image of uncropped blots and marker size are provided in Supplementary Fig. 1.

## AMPD activity

AMPD activity was analyzed by measuring the amount of ammonia released in response to AMP[68]. Fresh liver extracts (50 mg) were collected in a buffer containing 150 mM of KCl, 20 mM of Tris-HCl, pH 7.5, 1 mM of EDTA and 1 mM of dithiothreitol. The reaction mixture consisted of 25 mM of sodium citrate, pH 6.0, 50 mM of potassium chloride and varying concentrations of AMP. The enzyme reaction was initiated by the addition of the enzyme solution and incubated at 37 °C for 15 min for all samples collected. The reactions were stopped by adding the phenol/hypochlorite reagents: reagent A (100 mM of phenol and 0.050 g l$^{-1}$ sodium nitroprusside in water) was added, followed by reagent B (125 mM of sodium hydroxide, 200 mM of dibasic sodium phosphate and 0.1% sodium hypochlorite in $H_2O$). After incubation for 30 min at 25 °C, the absorbance of the samples was measured at 625 nm with a spectrophotometer. To determine ammonia production, a calibration curve was determined in the range of 5 μM to 1 mM of ammonia. Total ammonia was divided by 15 -number of minutes of the assay- and data presented as μmol ammonia produced per minute.

## Biochemical and tissue analysis

Blood was collected in Microtainer tubes (BD) from cardiac puncture of mice under isoflurane, and serum was obtained after centrifugation at 13,000 rpm for 2 min at room temperature. Serum and urine parameters were performed biochemically following the manufacturer's instructions: uric acid: DIUA-250, Phosphate: DIPI-500, ALT: EALT-100, AST: EAST-100 (Bioassay Systems Hayward, CA), Creatinine: C753291, Pointe Scientific. Fractional excretion of phosphate and uric acid was calculated by using the following ratio: FEx = (UX × PCr × 100)/(Px × UCr) where U and P refer to the urine and plasma concentrations of phosphate, uric acid and creatinine (Cr). Determination of parameters in tissue was performed in freeze-clamped tissues and measured biochemically following the manufacturer's protocol [uric acid: DIUA-250, Bioassay Systems; Hydroxyproline: DHYP-100, Bioassay Systems; thiobarbituric acid-reactive substances (TBARS): DTBA-100, Bioassay Systems; triglycerides (ETGA-200) and uric acid: DIUA-250, Bioassay Systems]. Triglyceride determination in liver was

analyzed in 5% triton-X homogenized samples and values normalized by soluble protein determined by BCA assay. To extract metabolites from liver samples, frozen liver samples were ground at liquid nitrogen temperature with a Cryomill (Retsch). The resulting tissue powder was weighed (~ 20 mg). The extraction was then done by adding –20 °C extraction solvent to the powder and incubating it at –20 °C overnight, followed by vortexing and centrifugation at 16,000 g for 10 min at 4 °C. The volume of the extraction solution (μl) was 40 times the weight of tissue (mg) to make an extract of 25 mg tissue per milliliter of solvent. Dried extracts were then redissolved in LC-MS Grade water (catalog 51,140; Thermo Fisher). Metabolites were analyzed via reverse-phase ion-pairing chromatography coupled to an Exactive Orbitrap mass spectrometer (Thermo Fisher Scientific). The mass spectrometer was operated in negative-ion mode with a resolving power of 100,000 at a mass-to-charge ratio (m/z) of 200 and a scan range of 75–1000 m/z. Energy charge was calculated as ([ATP] + 1/2[ADP])/[ATP] + [ADP] + [AMP][69,70].

## Statistics and reproducibility

All numeric data are presented as means ± SEM. Independent animal replicates for each data point ($n = 5$–8 mice per group) are shown in the figures. In general and to ensure replicability, body weight, fasting glucose and plasma ALT values were balanced across groups before the start of the studies. All animals were included in the analysis. Data graphics and statistical analysis were performed using Prism 5 (GraphPad). Data without indications were analyzed by one-way ANOVA with a Tukey post hoc test. $p$ values of $<0.05$ were regarded as statistically significant.

## Inclusion and diversity

One or more of the authors of this paper self-identifies as an underrepresented ethnic minority in science. While citing references scientifically relevant for this work, we also actively worked to promote gender balance in our reference list. The author list of this paper includes contributors from the location where the research was conducted who participated in the data collection, design, analysis, and/or interpretation of the work.

## Reporting summary

Further information on research design is available in the Nature Portfolio Reporting Summary linked to this article.

## Data availability

Further information and requests for resources and reagents should be directed and will be fulfilled by the corresponding author, Miguel A. Lanaspa (Miguel.lanaspagarcia@cuanschutz.edu). Mouse lines generated in this study are available for any researcher upon reasonable request. Similarly, all data reported in this paper will be shared by the corresponding author upon request. This paper does not report original code. Any additional information required to reanalyze the data reported in this paper is available from the corresponding author upon request. Raw source data for graphs/charts can be found in the Supplementary Data 1.

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

## Acknowledgements
This works was supported by NIH/NIDDK grants R01 DK121496 and DK108859 (M.A.L.).

## Author contributions
R.J.J., D.R.T. and M.A.L. designed the research; A.A.-H., D.J.O., M.K., M.F., R.J.J. D.R.T., and M.A.L. analyzed the data; A.A.-H., M.K., and M.A.L. performed the research; A.A.-H., and M.A.L. wrote the paper. All authors have read and agreed to the published version of the manuscript.

## Competing interests
The authors declare the following competing interests: R.J.J. and M.A.L. have several patents and patent applications related to blocking fructose metabolism in the treatment of metabolic diseases. R.J.J, M.A.L. and D.R.T. are also members of Colorado Research Partners LLC, that is developing inhibitors of fructose metabolism. R.J.J. also has some shares with XORT therapeutics, which is a startup company developing novel xanthine oxidase inhibitors. All other authors declare no competing interests.
