## [Peer Review File · Communications Biology]

Reviewers' comments:

Reviewer #1 (Remarks to the Author):

COMMSBIO-23-2409-T

"Activation of side-chain AMPD2 drives metabolic dysregulation and liver disease in mice with hereditary fructose intolerance."

Dr Lanaspá, et al. tried to identify metabolic pathways activated in hereditary fructose intolerance (HFI) associated with the progression of the disease by using aldolase B deficient mice. They described that the accumulation of unmetabolized fructose-1-phosphate activates a side-chain reaction, the purine degradation pathway, a metabolic route initiated by AMPD2 that regulates energy balance. They found that low amounts of fructose are sufficient to activate AMPD2 in HFI via a phosphate trap. Further, they described that deletion of AMPD2 in aldolase B deficient mice, corrects metabolic dysregulation, mobilizes fat and glycogen and ameliorates nonalcoholic steatohepatitis and liver disease while significantly increasing the tolerance of mice for fructose. Thus, they concluded that they found evidence of a new important metabolic pathway triggered by fructose metabolism that could be targeted to ameliorate metabolic dysfunction in HFI.

Major comments

Although AMPD2 is the isoform of AMP deaminase almost exclusively expressed in liver, but other tissues expressed AMPD3 or AMPD1 in addition to AMPD2 as the AMPD enzyme. Therefore, these other genes should be considered as well. Also, 5' nucleotidase and adenosine deaminase will affect the energy charge level in the purine degradation pathway. These genes and protein function should be discussed as well.

Minor comments

They stated that functional deletion of AMPD2 is demonstrated by reduced accumulation of inosine monophosphate (IMP) and the elevation of the AMP/IMP ratio dose-dependently (wild type vs AMPD2 heterozygous vs AMPD2 homozygous, Fig. 2D). AMPD enzyme activity showed dose dependency, though it seems that dose dependency of the AMP/IMP ratio was not apparent. (Page 7)

Reviewer #2 (Remarks to the Author):

Review of Manuscript# COMMSBIO-23-2409-T

Title: Activation of side-chain AMPD2 drives metabolic dysregulation and liver disease in mice with hereditary fructose intolerance.

Brief summary of the manuscript:

The authors claim that many, but not all, of the metabolic dysregulation seen in aldolase B-deficient

(aldoB^{-/-}) mice are due to the over-activation of AMPD2 and the excess production of uric acid. The involvement of AMPD2 and the purine degradation pathway in some of the metabolic dysregulations seen in aldoB^{-/-} mice is supported by the comparative increase in AMPD2 activity in these mice compared to WT mice. Additionally, the aldoB-AMPD2^{-/-} (double knock-out, DKO) has decreased [IMP], an increased AMP/IMP ratio, energy charge, and AMP/ATP, and phosphorylation of AMPK. The DKO shows reduced glycogen storage and triglyceride content in livers. Triglyceride content in the liver is a marker of fatty liver disease, seen in both Hereditary Fructose Intolerance patients (aldolase B deficient) and non-alcoholic fatty liver disease (aldolase B sufficient). Reduction of glycogen storage is implied to be a consequence of increased ability to utilize glycogen for energy within the liver or maintaining blood glucose levels during hypoglycemia. However, the deletion of AMPD2 does not prevent hypoglycemia with fructose exposure in aldoB^{-/-} or DKO mice.

The finding that AMPD2 is involved in purine degradation that leads to an increase in uric acid in response to fructose metabolism is novel. This supports previous findings that uric acid increases with excess fructose metabolism, that uric acid induces lipogenesis seen in fatty liver disease in NAFLD and HFI patients, and that inhibition of uric acid production with xanthine oxidase inhibitors reduces severity of fatty liver disease.

Increased AMPD2 activity in aldoB^{-/-} mice with or without fructose exposure is interesting. The increase in AMPD2 provides a mechanism for the increase in uric acid production in response to fructose metabolism, as well as a mechanism for the failure to activate AMPK. AMPK is activated by a change in the AMP/ATP ratio. Hepatic [ATP] decreases as ketohexokinase (KHK) phosphorylates fructose to produce fructose-1P (F1P), which cannot be cleaved without aldolase B activity; this sequestering of phosphate on a metabolite creates a “phosphate trap”. Presumably, under normal conditions, as [ATP] drops, AMPK should be phosphorylated in an attempt to increase ATP levels by activating kinases involved in glycogen release and fat metabolism. If AMPD2 activity metabolizes AMP to IMP (and eventually uric acid) fast enough, the AMP/ATP will stay constant and prevent AMPK activation.

While this manuscript presents findings that add to the current understanding of fatty liver disease in aldoB^{-/-} and these findings are of interest to NAFLD patients as well as HFI patients, there are several technical and conceptual flaws in the manuscript's current form.

1. There are significant and multiple grammatical errors that reduce readability. Many paragraphs are convoluted and contain multiple topics that would be better understood if they were broken into multiple paragraphs. There are multiple statements that are repetitive of each other and make it confusing as to whether the authors meant to present new information, or mistakenly repeated themselves. There are inconsistencies between using “fructokinase” vs “ketohexokinase”, which are the same enzyme. Aldolase “B” should be capitalized when talking about the protein. Genotypes should give the gene name and genotype (ex: aldoB^{-/-}). The document should be checked for spelling errors. There are also undefined acronyms such as WHO and AHA (line 46).

2. The abstract makes the claim that “deletion of AMPD2 in aldolase B deficient mice, corrects metabolic dysregulation, mobilizes fat and glycogen, and ameliorates nonalcoholic steatohepatitis and liver disease while significantly increasing the tolerance of mice for fructose.” This is an over-extrapolation of the

findings. AMPD2 deletion does increase the tolerance towards dietary fructose in DKO mice compared to *aldoB*^{-/-} mice when mice are allowed to eat or drink of their own volition. AMPD2 deletion does not increase the tolerance of fructose after an oral gavage, and in fact, glycogen stores may not be mobilized in DKO mice since they suffer from hypoglycemia to the same extent as *aldoB*^{-/-} mice do.

The conclusion that AMPD2 deletion leads to an increase in glycogen mobilization comes from the decrease in glycogen storage, and the increase in phosphorylated GS, in livers of DKO mice compared to *aldoB*^{-/-} mice when exposed to fructose. It is possible that glycogen stores are reduced in DKO because less glycogen is made in DKO mice compared to *aldoB*^{-/-} mice, and that glycogen levels are lower after fructose exposure because they were also lower before fructose exposure. The severe hypoglycemia in DKO mice with fructose gavage calls into question the ability to mobilize glycogen.

3. References are missing for some claims in the Introduction and Discussion:

- Line 45 mentions that HFCS in the current diet has increased exponentially since 1970, but does not state what the levels of fructose, sucrose, or HFCS were in 1970.
- Line 59 states that *aldoB*^{+/-} parents are “unaware sufferers” but does not give any information about what condition heterozygous parents suffer from that they are unaware of. If the authors meant that they are unaware carriers of the *aldoB*⁻ allele, this should be clarified. If the authors meant that heterozygous individuals develop liver (or kidney) problems at a higher rate than the general population, this should be clarified and cited.
- Line 66 mentions that “many” HFI patients develop cirrhosis or chronic kidney disease but does state how many HFI patients this is, or if cirrhosis and chronic kidney disease are over-represented in the HFI patient population compared to the non-HFI population.
- Line 70 states that “nothing is known about the sub-clinical and/or long-term effects from loss of aldolase B activity” when in fact, there are studies that have reported that *aldoB*^{+/-} family members are at a higher risk of developing type II diabetes, dietary fructose increases serum uric acid levels (2021 Debray et al., Clinical Nutrition), and that reduced aldolase B expression has been reported in cases of familial gout (1990 Seegmiller et al., PNAS).
- Line 70 also states that *aldoB*^{+/-} make up 2% of the entire population. Is this the global population, or the US population? There is not enough information about the *aldoB*⁻ allele frequency in many parts of the world to validate 2% of the “entire” population.
- Lines 321-322: There is no citation for the statement that HFI patients and *aldoB*^{-/-} mice are not able to fully convert fructose to fat.

4. Details are missing from the Methods section:

- Line 81 states that the animal protocol was approved by IACUC but does not give the IACUC number.
- There are details missing about how *aldoB*^{-/-} and the *aldoB*-AMPD2^{-/-} mice were bred and genotyped. It is not clear if the breeding pairs were fed fructose (or sucrose)-free chow and what chow pups were weaned on to before experiments.
- There should also be more information given about the macronutrient content of the various diets used. *aldoB*^{-/-} mice cannot cleave F1P, but they also cannot cleave FBP from glycolysis. The total carbohydrate content is important when evaluating the effects of different diets on liver metabolism and physiology.
- Line 94 states that male mice were used in this study but does not give a reason why female mice were

excluded.

- Lines 111-116 give an incomplete description of how livers were scored. The figures presented are at a magnification that is too low to see megamitochondria or mitotic figures, pigmented macrophages are not pointed out and there are no stains done to confirm macrophage cell markers, nor are there stains/assays done to confirm glycogenated nuclei, necrotic cells, or other features mentioned. If histological differences in H&E staining are used to assess all of these criteria, those should be shown. Additionally, the authors state that “we” obtained an average liver injury score across 10 images/animal but do not say how many individuals scored the animals, how the people scoring the images reconciled their liver scores, or if the people scoring the images were blinded to the animal/group identities. These details are important for reproducibility of reported findings.

- Line 92 mentions that “sugar water” was used but not what the final concentration of fructose, sucrose, or HFCS was used.

- Line 96 gives “Prior experience with ‘the model’ “ as a validation for not performing a priori power analysis, but does not state which model out of the three that were used in these experiments (aldoB^{-/-}, AMPD2^{-/-}, or the DKO). If the DKO was made as a novel mouse line for these studies, there would presumably be limited experience in working with this strain. The authors do not state what determines therapeutic efficacy for the aldoB^{-/-} mouse, or how many mice were used in each study.

- Line 107 mentions making composite images “PSR” staining, but does not define PSR staining or describe how PSR was visualized under polarized light. Picro-Sirius red staining for collagen is semi-quantitative and should be corroborated with another measure of collagen, such as the hydroxyproline assay for collagen quantification.

- Line 143 describes the determination of the absolute specific activity of ammonia production, but the Methods describe an end-point assay and a standard curve. It is not clear how rate was determined from this description.

- Line 169 says that all statistical analyses are presented as +/- SE. Is this meant to be “SD” for standard deviation or “SEM” for standard error of the mean? In the description of histological scores, it says that scores are presented as +/- standard deviation. In the legends for all figures, it says that error bars represent SEM.

There is insufficient detail, a lack of clarity, and sometimes contradictory statements in the Methods. Other researchers would not be able to repeat this work given the level of detail currently in this manuscript.

5. There are some logical oversights and an over-extrapolation of the Results based on the findings presented here.

- Line 176 states that the majority of fructose is metabolized by the liver and gut. In aldolase B-sufficient animals, fructose is metabolized by intestinal epithelial cells up to a specific threshold; it is not a linear relationship. There may be a dietary fructose bolus that overwhelms the epithelial cell capacity, at which point “fructose spillover” brings fructose to the liver. Whether or not the gut capacity metabolizes the “majority” of dietary fructose, or if the spillover to the liver is the majority, is concentration and time-dependent. Additionally, aldoB^{-/-} mice do not express aldolase B in epithelial intestinal, and none of the experiments presented look at the effects of F1P accumulation in the gut of aldoB^{-/-} or the DKO mice.

- Line 185 states that the upregulation of AMPD activity is exacerbated in aldoB^{-/-} mice exposed to dietary fructose, but not how long. The legend for Fig1 says that mice were kept on this diet for 2wks –

this should also be clear in the Methods and the text of the Results.

- Line 186: The authors posit that the increase in AMPD activity and decrease in cellular energy state paralleled each other. This implies that AMPD activity and cellular energy states were changing to the same degree, over time. It looks like all assays were done after 2wks of feeding, which would mean that there were positively correlated with each other at one time point, but not necessarily that the change in those two variables was parallel over the 2wks. A time course needs to be done in order to say that these two variables change in a way that parallels each other over time.

- Line 198 describes decreased ACC phosphorylation in *aldoB*^{-/-} mice compared to WT mice exposed to fructose. The authors state that decreased pACC may lead to increased fat accumulation with dietary fructose. However, the difference in fat accumulation and inflammation between *aldoB*^{-/-} mice and WT mice both exposed to fructose is not an appropriate control for making this statement; instead, *aldoB*^{-/-} mice with or without dietary fructose should be compared. It is possible that *aldoB*^{-/-} mice have reduced pACC and increased fat accumulation without dietary fructose.

- Line 202 states that “F1P is being recognized as a very important molecule driving nutrient uptake...” but later in the manuscript, the authors show that increased F1P accumulation in *aldoB*^{-/-} mice fed fructose reduces feeding behavior. The current wording implies that F1P increases feeding, which is in contrast with the results presented.

- Line 205 describes the experimental rationale for making the *aldoB*-AMPD2^{-/-} double-knockout (DKO) mice as an attempt to separate the effects of an increase in F1P from the effects of an increase in AMPD2 activity. To understand the role of F1P in activation of AMPD2, *aldoB*^{-/-} mice fed a fructose- (or sucrose)-free diet should be compared to *aldoB*^{-/-} mice exposed to fructose. The role of F1P cannot be properly evaluated apart from AMPD2 activation, or other potential metabolic disturbances, without comparison to the DKO with lower F1P levels. Alternatively, fructose-fed AMPD2^{-/-} mice that are *aldoB*^{+/-} could be compared to AMPD2^{-/-} and the DKO to see if there is a dose-dependent effect of F1P accumulation.

- Line 212: It is not clear from the nomenclature used to describe the mice whether the AMPD2 homozygous are AMPD2^{-/-} (KO homozygous) or ^{+/+} (WT homozygous), although the former is implied since WT mice are used as a control group. It is also not clear whether these are meant to be double-KO mice. Fig2 B does not clarify if these mice are DKOs; DKOs are implied but the text says “AMPD2 heterozygous (KH) and “AMPD2 homozygous (KK)”. The choice of nomenclature for the different strains is not standard.

- Lines 216-218: The authors note that DKO mice activate AMPK before exposure to fructose. This points to a pathway that is mitigating fatty liver in the DKO mice that is separate from the accumulation of F1P. AMPD2 deletion is not protecting these mice from damage induced by fructose-feeding; it is protecting them from damage induced by general metabolic imbalance, which then makes fructose-dependent damage less severe.

- Line 243: The authors provide evidence that the DKO mice consume more dietary fructose than *aldoB*^{-/-} mice and state that this “indicates a marked improvement in the tolerance of the (DKO) mice to fructose”. However, in the section starting on line 268, fructose gavage induces severe hypoglycemia in both *aldoB*^{-/-} and DKO mice. It is an over-extrapolation to say that increased feeding in the DKO mice improves fructose tolerance since fructose is not tolerated better by DKO mice in all circumstances. The concentration-dependent, and possible time-dependent, influences of when fructose is better tolerated are important to note.

- Line 275 claims that AMPD2 deletion blocks the degradation of purines and does not change the accumulation of F1P in HFI patients. Given that there are some physiological differences between mice

and humans, the results in *aldoB*^{-/-} mice need to be confirmed in human HFI patients before this statement can be made.

- Line 290 mentions nutrients are lost in the urine but does not state which nutrients. Fig3G and 3H show the excretion of phosphate and glucose; this should be made explicit in the text. The authors should also address why phosphate might be lost in the urine if it is being sequestered inside hepatocytes due to getting trapped on F1P.

- The schematic presented in Fig1A implies that F1P directly interacts with AMPD2. There is no evidence that this interaction is direct. This schematic also shows ATP being converted to AMP by KHK when fructose is phosphorylated to F1P, and this is not accurate. ATP is converted to ADP by KHK; conversion of ADP to AMP happens in other reactions.

- In the text, the authors claim that uric acid is ultimately responsible for some of the negative physiological consequences of fructose metabolism in *aldoB*^{-/-} mice. In Fig 3K, *aldoB*^{-/-} mice exposed to either dietary fructose or dietary OxAc have similar amounts of uric acid. However, in Fig 3L, *aldoB*^{-/-} mice that are fed fructose alone, or fed fructose with OxAc, have similar decreases in body weight, while *aldoB*^{-/-} mice fed OxAc alone, but not fructose, do not experience the same decrease in body weight as those fed fructose. This points to a mechanism distinct from uric acid production or accumulation that drives pathology in *aldoB*^{-/-} mice.

6. There are important controls missing that are needed to support the conclusions drawn in this manuscript.

- If F1P is the driving factor that increases AMPD2 activity and the purine degradation pathway, then *aldoB*^{-/-} mice that are not fed fructose need to be shown as a control for the liver pathology scores and improvement seen in the DKO mice. If *aldoB*^{-/-} mice on a fructose-free diet benefit to the same degree, then F1P is not responsible for activating AMPD2 activity.

- If AMPD2 deletion activates GS and allows glycogen to be mobilized to restore energy balance after fructose exposure, then glycogen levels in the DKO mice need to be measured before fructose exposure.

- There are no control images shown to support the claim that fructose induces kidney damage in *aldoB*^{-/-} mice. It may be that *aldoB*^{-/-} mice not fed fructose have similar kidney damage.

While the results presented here are striking, the conclusions drawn are not fully supported by those results. Limitations of this study and alternative hypotheses need to be addressed in the manuscript.

Reviewer #3 (Remarks to the Author):

In the current study, the authors Andres-Hernando et al, investigated the metabolic disorders caused by

inefficient aldolase B. The study performed in mice that were bred with pure C57BL/6J to obtain both knockout and WT littermates and further crossed with aldolase B KO mice to obtain aldolase b/AMPD2 double KO mice. The mice allowed ad libitum access to 1 % fructose or 2.5 % oxonic acid chow (w/w), compared to fructose-free diet. The authors used histopathology, western blotting, AMPD activity and tissue analysis of various metabolites in order to identify the metabolic pathways activated in hereditary fructose intolerance associated with the progression of the disease.

Results described by the authors including:

- (i) The accumulation of fructose-1-phosphate causes a side-chain reaction including, the purine degradation pathway and a metabolic route initiated by AMPD2 that regulates energy balance.
- (ii) Low amounts of fructose are sufficient to activate AMPD2 in HFI via a phosphate trap
- (iii) Loss of AMPD2 in aldolase B deficient mice, corrects metabolic malfunction and amends nonalcoholic steatohepatitis and liver disease while increasing the tolerance for fructose.

The authors concluded that this manuscript reveals new important metabolic pathways triggered by fructose metabolism that can be targeted to improve metabolic dysfunction in HFI.

A major concern is the lack of depth and granularity in the experimental design and presentation of the data. The following are highly recommended:

1. Regarding experimental design, it is very important to identify the function and dysfunction of a specific metabolic pathway by using stable isotope tracer, such as ¹³C-labeled fructose given by oral gavage or IV infusion. The authors can trace the exact metabolism of fructose determine specific activity, such as flux rate and determining the site of malfunction. Examples of such investigation can be seen in: *Biochem J.* 2012 Jun 15;444(3):537-51; *Am J Sports Med* 2018 Jul;46(9):2222-2231; *Cell Metab.* 2020 Jan 7; 31(1): 174–188.e7.

2. To specifically determine the liver metabolic malfunction in WT versus aldolase B KO and aldolase b/AMPD2 double knockout, it is prefer using liver perfusion experiment with ¹³C-labeled fructose or another precursor. The in vivo study as presented in the current paper cannot specifically differentiate between metabolism in the liver and the other organs in the body.

3. It is highly recommended that the authors provide a companion figure, e.g., heatmap to show the markers used to defining the metabolic disorder and to help navigate the identity of a specific metabolic site and enable comparison to the overall metabolic function (data in Fig 1 D-G, Fig 2 C-G, M, N, Fig 3, A-H).

Minor issues including inconsistency in the text presentation and numerous spelling mistakes that need to be corrected.

Overall response: We would like to thank the reviewers for their positive comments and for finding our manuscript “novel” and “of interest”. In this revised version we have aimed to address all the concerns raised by the reviewers which we believe have resulted in a much improved manuscript. An specific point-by-point response is included below:

Reviewer #1 (Remarks to the Author):

COMMSBIO-23-2409-T

"Activation of side-chain AMPD2 drives metabolic dysregulation and liver disease in mice with hereditary fructose intolerance."

Dr Lanaspá, et al. tried to identify metabolic pathways activated in hereditary fructose intolerance (HFI) associated with the progression of the disease by using aldolase B deficient mice. They described that the accumulation of unmetabolized fructose-1-phosphate activates a side-chain reaction, the purine degradation pathway, a metabolic route initiated by AMPD2 that regulates energy balance. They found that low amounts of fructose are sufficient to activate AMPD2 in HFI via a phosphate trap. Further, they described that deletion of AMPD2 in aldolase B deficient mice, corrects metabolic dysregulation, mobilizes fat and glycogen and ameliorates nonalcoholic steatohepatitis and liver disease while significantly increasing the tolerance of mice for fructose. Thus, they concluded that they found evidence of a new important metabolic pathway triggered by fructose metabolism that could be targeted to ameliorate metabolic dysfunction in HFI.

Major comments

Although AMPD2 is the isoform of AMP deaminase almost exclusively expressed in liver, but other tissues expressed AMPD3 or AMPD1 in addition to AMPD2 as the AMPD enzyme. Therefore, these other genes should be considered as well. Also, 5'nucleotidase and adenosine deaminase will affect the energy charge level in the purine degradation pathway. These genes and protein function should be discussed as well.

Response: We thank the reviewer for the insight. The reviewer is correct in that other isoforms of AMPD besides AMPD2 may play an important role in the overall phenotype observed in fructose exposed aldob(-/-) mice. We have focused on AMPD2 as the main liver isoform for the following reasons: 1) the majority of dietary fructose is metabolized in gut and liver which primarily express AMPD2. 2) AMPD1 is a muscle-specific isoform of AMPD. Fructose metabolism in muscle is minimal and no fructose transporters (glut5.-hepatocytes, small intestine and kidney- glut8 -testis and perhaps adipose-- and glut2 -hepatocytes, small intestine and kidney-) are expressed in the skeletal muscle 3) AldoB expression is limited to liver, small intestine and kidney (<https://www.proteinatlas.org/ENSG00000136872-ALDOB/tissue>). In consequence, these organs are the ones primarily affected in HFI. and 4) There are cell-type specific differences in the liver between AMPD2 and AMPD3. AMPD2 is the hepatocyte-specific isoform of AMPD while AMPD3 is mostly present in myeloid-derived cells. Further, our new data (Figure 4) in cell-specific AMPD2 deficient mice demonstrate that hepatocyte AMPD2 drives HFI in aldob(-/-) mice.

As suggested, a paragraph discussing the potential involvement in HFI of other enzymes related with the purine degradation pathway has been added. Not much is known on the potential role of adenosine in HFI. However, adenosine, produced from AMP under conditions of low energy charge (hypoxia, inflammation) elicits cytoprotective actions in response to the low energy state in many tissues including the liver. Our data suggests that the energy charge in HFI is particularly low and thus adenosine may be important to protect cells from energy depletion. Therefore, excessive elimination of adenosine via ADA may exert important deleterious consequences in HFI. Further, adenosine metabolism via ADA increases the production of metabolites in the purine degradation pathway like inosine which in turn thus contributing to the production and accumulation of uric acid characteristic of human subjects with HFI.

Minor comments

They stated that functional deletion of AMPD2 is demonstrated by reduced accumulation of inosine

monophosphate (IMP) and the elevation of the AMP/IMP ratio dose-dependently (wild type vs AMPD2 heterozygous vs AMPD2 homozygous, Fig. 2D). AMPD enzyme activity showed dose dependency, though it seems that dose dependency of the AMP/IMP ratio was not apparent. (Page 7).

Response: It is correct that the elevation of the AMP/IMP ratio is more prominent in AMPD2 homozygous and the intermediate effect in AMPD2 heterozygous seems closer to wild types than knockouts. However, raw data analysis shows a clear up-regulation of the AMP/IMP ratio in AMPD2 hets (52-fold) compared to wild types (0.29 ± 0.08 in wild type, vs 15.28 ± 11.38 in AMPD2 hets vs 142.8 ± 65.05 in AMPD2 knockouts) indicating a dose-dependent effect of AMPD2 loss in the AMP/IMP ratio.

Reviewer #2 (Remarks to the Author):

Brief summary of the manuscript:

The authors claim that many, but not all, of the metabolic dysregulation seen in aldolase B-deficient (aldob^{-/-}) mice are due to the over-activation of AMPD2 and the excess production of uric acid. The involvement of AMPD2 and the purine degradation pathway in some of the metabolic dysregulations seen in aldob^{-/-} mice is supported by the comparative increase in AMPD2 activity in these mice compared to WT mice. Additionally, the aldob-AMPD2^{-/-} (double knock-out, DKO) has decreased [IMP], an increased AMP/IMP ratio, energy charge, and AMP/ATP, and phosphorylation of AMPK. The DKO shows reduced glycogen storage and triglyceride content in livers. Triglyceride content in the liver is a marker of fatty liver disease, seen in both Hereditary Fructose Intolerance patients (aldolase B deficient) and non-alcoholic fatty liver disease (aldolase B sufficient). Reduction of glycogen storage is implied to be a consequence of increased ability to utilize glycogen for energy within the liver or maintaining blood glucose levels during hypoglycemia. However, the deletion of AMPD2 does not prevent hypoglycemia with fructose exposure in aldob^{-/-} or DKO mice.

The finding that AMPD2 is involved in purine degradation that leads to an increase in uric acid in response to fructose metabolism is novel. This supports previous findings that uric acid increases with excess fructose metabolism, that uric acid induces lipogenesis seen in fatty liver disease in NAFLD and HFI patients, and that inhibition of uric acid production with xanthine oxidase inhibitors reduces severity of fatty liver disease.

Increased AMPD2 activity in aldob^{-/-} mice with or without fructose exposure is interesting. The increase in AMPD2 provides a mechanism for the increase in uric acid production in response to fructose metabolism, as well as a mechanism for the failure to activate AMPK. AMPK is activated by a change in the AMP/ATP ratio. Hepatic [ATP] decreases as ketohexokinase (KHK) phosphorylates fructose to produce fructose-1P (F1P), which cannot be cleaved without aldolase B activity; this sequestering of phosphate on a metabolite creates a “phosphate trap”. Presumably, under normal conditions, as [ATP] drops, AMPK should be phosphorylated in an attempt to increase ATP levels by activating kinases involved in glycogen release and fat metabolism. If AMPD2 activity metabolizes AMP to IMP (and eventually uric acid) fast enough, the AMP/ATP will stay constant and prevent AMPK activation.

While this manuscript presents findings that add to the current understanding of fatty liver disease in aldob^{-/-} and these findings are of interest to NAFLD patients as well as HFI patients, there are several technical and conceptual flaws in the manuscript's current form.

Response: Thanks, this is a great summary of the evidence we present in our manuscript. We understand the reviewers technical and conceptual concerns and thus, we have tried to fully address them in the revised version.

1.

1a) *There are significant and multiple grammatical errors that reduce readability. Many paragraphs are convoluted and contain multiple topics that would be better understood if they were broken into multiple paragraphs. There are multiple statements that are repetitive of each other and make it confusing as to whether the authors meant to present new information, or mistakenly repeated themselves. There are inconsistencies between using “fructokinase” vs “ketohexokinase”, which are the same enzyme. Aldolase “B” should be capitalized when talking about the protein. Genotypes should give the gene name and genotype (ex: aldob^{-/-}).*

The document should be checked for spelling errors. There are also undefined acronyms such as WHO and AHA (line 46).

Response: We have thoroughly checked the text correcting typos and making it more readable. For aldolase B, we have consistently referenced it as aldob (mouse) or aldob (human) for the gene and aldolase B for the protein. For AMPD2, it is termed as ampd2 for the gene and AMPD2 for the protein. In total, we have made over 350 single edits to the text in order to substantially improve the flow and readability of the manuscript.

1b). The abstract makes the claim that “deletion of AMPD2 in aldolase B deficient mice, corrects metabolic dysregulation, mobilizes fat and glycogen, and ameliorates nonalcoholic steatohepatitis and liver disease while significantly increasing the tolerance of mice for fructose.” This is an over-extrapolation of the findings. AMPD2 deletion does increase the tolerance towards dietary fructose in DKO mice compared to aldob^{-/-} mice when mice are allowed to eat or drink of their own volition. AMPD2 deletion does not increase the tolerance of fructose after an oral gavage, and in fact, glycogen stores may not be mobilized in DKO mice since they suffer from hypoglycemia to the same extent as aldob^{-/-} mice do.

The conclusion that AMPD2 deletion leads to an increase in glycogen mobilization comes from the decrease in glycogen storage, and the increase in phosphorylated GS, in livers of DKO mice compared to aldob^{-/-} mice when exposed to fructose. It is possible that glycogen stores are reduced in DKO because less glycogen is made in DKO mice compared to aldob^{-/-} mice, and that glycogen levels are lower after fructose exposure because they were also lower before fructose exposure. The severe hypoglycemia in DKO mice with fructose gavage calls into question the ability to mobilize glycogen.

Response: We agree with the reviewer and have edited the abstract toning down the relevance of our findings. However, to the reviewer’s comment on glycogen mobilization in aldob^(-/-) and aldob/ampd2^(-/-) mice, we would like to point that we have previously demonstrated that the acute hypoglycemic effect in aldob^(-/-) is the consequence of enhanced glucose uptake by the liver and not dependent on insulin actions. Both insulin and glucagon levels are not elevated during the length of the fructose challenge. This is important because glycogen mobilization by hepatocytes is not a direct response to low glucose in the circulation but instead to the glycogenolytic consequences of glucagon secreted in response to hypoglycemia¹. Therefore, as the reviewer indicates, we cannot assume from the acute challenge neither that there is improved mobilization of glycogen in aldob/ampd2^(-/-) mice nor that there is still an impairment in glycogen mobilization in aldob/ampd2^(-/-) mice as glucagon is not elevated during the length challenge. As such, the statement regarding glycogen mobilization has been edited in the abstract and text to better reflect our findings: that at baseline, glycogen levels in aldob/ampd2^(-/-) are not significantly different from wild type mice (Figure 2I) and significantly lower than in aldob^{-/-} mice suggesting that in aldob/ampd2^(-/-) mice there is not a defect in glycogen accumulation as it exists in aldob^{-/-} mice.

. References are missing for some claims in the Introduction and Discussion:
1c) - Line 45 mentions that HFCS in the current diet has increased exponentially since 1970, but does not state what the levels of fructose, sucrose, or HFCS were in 1970.

Response: Corrected

1d) - Line 59 states that aldob^{+/-} parents are “unaware sufferers” but does not give any information about what condition heterozygous parents suffer from that they are unaware of. If the authors meant that they are unaware carriers of the aldob⁻ allele, this should be clarified. If the authors meant that heterozygous individuals develop liver (or kidney) problems at a higher rate than the general population, this should be clarified and cited.

Response: Thanks for noticing. Heterozygous individuals do not present clinical manifestations of the disease. We meant to indicate that because of this, parents that are heterozygous are often unaware carriers of the aldob⁻ allele and therefore, weaning off breastmilk and transitioning to sugar-containing foods is one of the greatest risk periods for the condition to be manifested clinically. We have edited the sentence accordingly to clarify its meaning.

1e) - Line 66 mentions that “many” HFI patients develop cirrhosis or chronic kidney disease but does state how many HFI patients this is, or if cirrhosis and chronic kidney disease are over-represented in the HFI patient population compared to the non-HFI population.

Response: Unfortunately, there not large clinical studies analyzing long-term consequences of HFI and we rely in small studies with often less than 15-20 individuals. The data however, points to the high incidence of both liver and kidney disease in the HFI community. For example, recently, in a small cohort study from well controlled patients, Aldamiz-Echevarria et al have reported that over 60% of the subjects analyzed had non-alcoholic fatty liver disease². Similarly, the effect of HFI in kidney disease is known since the 1960s³. In this regard, Simons et al, recently reported a higher prevalence of high blood pressure -a consequence of renal dysfunction- in subjects with HFI⁴. We have edited the text providing references to support the claim that the prevalence of kidney and liver disease is substantially high in HFI individuals.

1f) - Line 70 states that “nothing is known about the sub-clinical and/or long-term effects from loss of aldolase B activity” when in fact, there are studies that have reported that aldob⁺/⁻ family members are at a higher risk of developing type II diabetes, dietary fructose increases serum uric acid levels (2021 Debray et al., *Clinical Nutrition*), and that reduced aldolase B expression has been reported in cases of familial gout (1990 Seegmiller et al., *PNAS*).

Response: We thank the reviewer for bringing those studies which are now included in the paragraph which we have edited accordingly. We were aware of the recent Debray paper, it is very well designed and constructed and to our knowledge it is one of the few analyzing the response of aldob- allele carriers.

1g) - Line 70 also states that aldob⁺/⁻ make up 2% of the entire population. Is this the global population, or the US population? There is not enough information about the aldob⁻/⁻ allele frequency in many parts of the world to validate 2% of the “entire” population.

Response: It is true that the worldwide prevalence of HFI or heterozygotes is unknown due to the difficulty of its diagnosis. First reports from Europe (Switzerland and the UK) in the 1970s and 1990s estimated an incidence of HFI of 1:20,000⁵. It is likely that the incidence rate varies quite widely among different ethnic groups. However, there have been numerous reports of self diagnosis in adulthood, inadvertent deaths to undiagnosed subjects, and homozygous-heterozygous marriages, all of which indicate that the incidence rate could be closer to 1 in 10,000. This would suggest that the carrier frequency would be 1 in 50. In this regard, coffee et al, recently estimated that the heterozygous allele population in the US would range between 1:50 and 1:125⁶. Because it is difficult to extrapolate from this data the allele distribution globally, we have edited the paragraph emphasizing that this is just an estimation of its prevalence,

1h) - Lines 321-322: There is no citation for the statement that HFI patients and aldob⁻/⁻ mice are not able to fully convert fructose to fat.

Response: The lipogenic effects of fructose are well characterized in the literature⁷. However, and as we mention in the next sentence only a small percentage (1-3%) of the fructose molecule is actually converted to fat. Presumably, and as shown in Figure 5A and in our JCI paper, since a high percentage of fructose stays as F1P in aldob(-/-) mice, the percentage of fructose that is directly converted to fat in aldob(-/-) individuals is even lower. This does not necessarily mean that there is not de novo lipogenesis induced by fructose and maximized in HFI, it means that fructose is not an important lipogenic substrate but rather it potentiates (or amplifies) the expression of lipogenic enzymes. Zhao et al⁸ proposed that fructose-derived acetate by the microbiota is the lipogenic substrate and we showed that liver fructose metabolism is necessary for the development of fatty liver disease⁹. Collectively, the current proposed model on the lipogenic effects of fructose is that hepatic fructose metabolism promotes the up-regulation of lipogenic enzymes (ACC, FAS, ACL) while acetate produced from fructose metabolism by the microbiota provides the lipogenic substrate. Interestingly, we published that the expression of ACC, ACL and FAS are up-regulated in fructose-exposed AldoB(-/-) mice¹⁰ and blocked when fructokinase expression is deleted. This suggest that the up-regulation of these lipogenic enzymes is mediated either by F1P itself or the activation of AMPD2 and the purine degradation side chain. Since aldob/ampd2(-/-) do not develop fatty liver disease, it likely suggests that the side chain driven by AMPD2 drives the hepatic-component of fatty liver disease induced by fructose.

2. Details are missing from the Methods section:

2a)- Line 81 states that the animal protocol was approved by IACUC but does not give the IACUC number.

Response: Included.

2b)- There are details missing about how aldob^{-/-} and the aldob-AMPD2^{-/-} mice were bred and genotyped. It is not clear if the breeding pairs were fed fructose (or sucrose)-free chow and what chow pups were weaned on to before experiments.

Response: We have expanded in the Methods, the details on the breeding strategy. In general, mice were not previously exposed to fructose as it could condition the response of the mice to the intervention. To this end, breeding pairs were also maintained in sucrose-free diet.

2c) - There should also be more information given about the macronutrient content of the various diets used. aldob^{-/-} mice cannot cleave F1P, but they also cannot cleave FBP from glycolysis. The total carbohydrate content is important when evaluating the effects of different diets on liver metabolism and physiology.

Response: Carbohydrate content (total and source) is now included in the methods. The reviewer is right in that glycolysis is impaired as FBP is not cleaved either. Further, in our characterization of aldob^{-/-} mice¹⁰, we showed that there were no compensation by other aldolases (Aldolase A and Aldolase C) to the deficiency of AldoB in the liver. However, unlike fructolysis, glycolytic metabolites can be provided by anaplerotic routes. Even though it is outside of the focus of the paper, we believe that the pentose phosphate route may be overactivated in response to aldob^{-/-} deficiency to provide from glucose-6-phosphate, metabolites downstream FBP including glyceraldehyde-3-phosphate that can support glycolysis and oxidative phosphorylation. Unfortunately, the metabolism of fructose is very dependent on fructokinase as the affinity of other hexose kinases for fructose is very low and as such the majority of fructose is rapidly metabolized via fructokinase.

2d) - Line 94 states that male mice were used in this study but does not give a reason why female mice were excluded.

Response: We apologize for the error that is now corrected. Both males and females were employed in the study. We have not observed sex differences in the response to fructose of aldob^{-/-} mice. Further, the breeding was complicated enough (only expected 12.5% of aldob/ampd2^{-/-} pups from aldob/ampd2^{+/-} breeding pairs that we could not afford to perform sex-specific studies.

2e) - Lines 111-116 give an incomplete description of how livers were scored. The figures presented are at a magnification that is too low to see megamitochondria or mitotic figures, pigmented macrophages are not pointed out and there are no stains done to confirm macrophage cell markers, nor are there stains/assays done to confirm glycogenated nuclei, necrotic cells, or other features mentioned. If histological differences in H&E staining are used to assess all of these criteria, those should be shown. Additionally, the authors state that “we” obtained an average liver injury score across 10 images/animal but do not say how many individuals scored the animals, how the people scoring the images reconciled their liver scores, or if the people scoring the images were blinded to the animal/group identities. These details are important for reproducibility of reported findings.

Response: As suggested, a better description of the liver score is now provided in the methodology. All animals in the experiment have been scored by our pathologist (Dr Orlicky) following our modified version of the Brunt scoring system¹⁰. References on how the scoring is tallied are now included. Number of mice per group are included in all figure legends.

2f) - Line 92 mentions that “sugar water” was used but not what the final concentration of fructose, sucrose, or HFCS was used.

Response: The concentration of sugar in the drinking water is now included in both the methods and figure legends (5% w/v).

2g) - Line 96 gives “Prior experience with ‘the model’ “ as a validation for not performing a priori power analysis, but does not state which model out of the three that were used in these experiments (aldob^{-/-}, AMPD2^{-/-}, or the DKO). If the DKO was made as a novel mouse line for these studies, there would presumably be limited experience in working with this strain. The authors do not state what determines therapeutic efficacy for the aldob^{-/-} mouse, or how many mice were used in each study.

Response: We have edited the text specifying the models we are referring to. Both the acute fructose challenge and the chronic exposure to fructose (1%) models have been characterized in the past by our lab in aldob^{-/-} mice^{10,11} and power analysis inferred from those previous studies. Number of mice in each study is indicated in the figure legends.

2h) - Line 107 mentions making composite images “PSR” staining, but does not define PSR staining or describe how PSR was visualized under polarized light. Picro-Sirius red staining for collagen is semi-quantitative and should be corroborated with another measure of collagen, such as the hydroxyproline assay for collagen quantification.

Response: PSR is now better described in the methods. Also, we are providing brightfield and polarized light images of the same representative sections and validated biochemically with hydroxyproline the histological findings.

2i) - Line 143 describes the determination of the absolute specific activity of ammonia production, but the Methods describe an end-point assay and a standard curve. It is not clear how rate was determined from this description.

Response: The standard curve was used to extrapolate the amount of ammonia present in the samples. To calculate the rate, the extrapolated values (in uM) were divided by 15 as it is the number of minutes in the assay. This is not a kinetic study to determine rate of appearance but rather a determination of total ammonia produced during the length of the assay (15 minutes). We have edited and corrected the text to better clarify the procedure.

2j) - Line 169 says that all statistical analyses are presented as +/- SE. Is this meant to be “SD” for standard deviation or “SEM” for standard error of the mean? In the description of histological scores, it says that scores are presented as +/- standard deviation. In the legends for all figures, it says that error bars represent SEM.

Response: All data is presented as SEM including histology analysis and liver score. The typo is now corrected.

2k) There is insufficient detail, a lack of clarity, and sometimes contradictory statements in the Methods. Other researchers would not be able to repeat this work given the level of detail currently in this manuscript.

Response: We thank the reviewer for their thorough scrutiny of the section. We have aimed to correct all the concerns and hopefully have helped clarify the methodology.

3. There are some logical oversights and an over-extrapolation of the Results based on the findings presented here.

3a) - Line 176 states that the majority of fructose is metabolized by the liver and gut. In aldolase B-sufficient animals, fructose is metabolized by intestinal epithelial cells up to a specific threshold; it is not a linear relationship. There may be a dietary fructose bolus that overwhelms the epithelial cell capacity, at which point “fructose spillover” brings fructose to the liver. Whether or not the gut capacity metabolizes the “majority” of dietary fructose, or if the spillover to the liver is the majority, is concentration and time-dependent. Additionally, aldob^{-/-} mice do not express aldolase B in epithelial intestinal, and none of the experiments presented look at the effects of F1P accumulation in the gut of aldob^{-/-} or the DKO mice.

Response: We agree with the reviewer but believe the statement is still correct as is. Dietary fructose is metabolized by both gut and liver. The reviewer is correct that lower concentrations are principally metabolized in the gut as Jang et al first demonstrated¹² and either high amounts, or chronic exposure to fructose increases

hepatic delivery of fructose. However, using tissue specific knockout for fructokinase, our group demonstrated that the specific metabolism of dietary fructose in the liver drives metabolic syndrome⁹. This observation has been further validated by Park et al using liver-specific KHK siRNAs¹³. The importance of the role of the liver in the metabolism of dietary fructose is even more relevant in mice with great sensitivity for fructose like the aldob(-/-) mice. Just for the reviewer, and outside the scope of this manuscript, we have conducted pilot studies in which the expression of fructokinase has been deleted in either the small intestine or liver of aldob(-/-). Similar to what we found in Andres-Hernando et al⁹, deletion of fructokinase in the liver completely protected aldob(-/-) against fructose-induced liver and kidney disease as well as acute hypoglycemia while its deletion in the gut exacerbated the phenotype by increasing the amount of fructose delivered to the liver. Collectively, we believe that these studies demonstrate the importance of both gut and liver in dietary fructose metabolism and therefore, we prefer to keep the statement in line 176 as is.

3b) - Line 185 states that the upregulation of AMPD activity is exacerbated in aldob(-/-) mice exposed to dietary fructose, but not how long. The legend for Fig1 says that mice were kept on this diet for 2wks – this should also be clear in the Methods and the text of the Results.

Response: AMPD activity was determined in mice with no prior exposure to fructose(baseline) and mice exposed to fructose for 2 weeks. We have edited line 185 specifying the timing in which AMPD activity was measured.

3c) - Line 186: The authors posit that the increase in AMPD activity and decrease in cellular energy state paralleled each other. This implies that AMPD activity and cellular energy states were changing to the same degree, over time. It looks like all assays were done after 2wks of feeding, which would mean that there were positively correlated with each other at one time point, but not necessarily that the change in those two variables was parallel over the 2wks. A time course needs to be done in order to say that these two variables change in a way that parallels each other over time.

Response: We understand the reasoning behind the reviewer's comment and agree with it. Therefore, we have edited the text substituting the term "paralleled" which implies a correlation over time between the two variables (AMPD activity and energy state) which we have not determined continuously to "associated" which does not necessarily imply a continuous correlation between the variables over time.

3d) - Line 198 describes decreased ACC phosphorylation in aldob(-/-) mice compared to WT mice exposed to fructose. The authors state that decreased pACC may lead to increased fat accumulation with dietary fructose. However, the difference in fat accumulation and inflammation between aldob(-/-) mice and WT mice both exposed to fructose is not an appropriate control for making this statement; instead, aldob(-/-) mice with or without dietary fructose should be compared. It is possible that aldob(-/-) mice have reduced pACC and increased fat accumulation without dietary fructose.

Response: We have included in Figure 1 (Fig 1L) liver triglyceride content from the four groups. As shown in the panel and consistent with the histology data, aldob(-/-) mice on fructose demonstrate the highest triglyceride content of all groups. Interestingly, aldob(-/-) mice without dietary fructose also have higher triglyceride content than wild type counterparts. This is consistent with a previous report from our group (Lanaspa et al¹⁰, Table 1). One possibility that may explain the increased fat accumulation without dietary fructose in aldob(-/-) mice, is the production and metabolism of endogenous fructose from glucose via the polyol pathway. In this regard, we have recently demonstrated that aldob(-/-) mice are sensitive not only to fructose but also foods that activate the polyol pathway and promote the endogenous production of fructose. These activators include alcohol, glucose solutions, as well as intermediates of the polyol pathway like sorbitol¹¹. As the sucrose-free chow is very rich in complex carbohydrates (50% dextrose and 12.9 % corn starch) it is likely there is a chronic partial conversion of glucose into endogenous fructose that would help explain the metabolic baseline differences observed in Figure 1 between wild type and aldob(-/-) mice without dietary fructose.

3e) - Line 202 states that "F1P is being recognized as a very important molecule driving nutrient uptake..." but later in the manuscript, the authors show that increased F1P accumulation in aldob(-/-) mice fed fructose reduces feeding behavior. The current wording implies that F1P increases feeding, which is in contrast with the results presented.

Response: We apologize for the confusion. The referenced manuscripts in that sentence demonstrate that F1P acts as a signaling molecule to promote intestinal hypoxic cell survival and nutrient absorption (Taylor et al), and lipid storage (Brouwers reference). However, these papers do not show increased feeding and we apologize if that was the understanding of the reviewer from the sentence in line 202. We have edited the sentence substituting “uptake” -intake- by absorption.

3f) - Line 205 describes the experimental rationale for making the aldob-AMPD2^{-/-} double-knockout (DKO) mice as an attempt to separate the effects of an increase in F1P from the effects of an increase in AMPD2 activity. To understand the role of F1P in activation of AMPD2, aldob^{-/-} mice fed a fructose- (or sucrose)-free diet should be compared to aldob^{-/-} mice exposed to fructose. The role of F1P cannot be properly evaluated apart from AMPD2 activation, or other potential metabolic disturbances, without comparison to the DKO with lower F1P levels. Alternatively, fructose-fed AMPD2^{-/-} mice that are aldob^{+/-} could be compared to AMPD2^{-/-} and the DKO to see if there is a dose-dependent effect of F1P accumulation.

Response: The comparison between aldob^(-/-) fed a sucrose-free diet with aldob^(-/-) exposed to fructose is already shown in Figure 1 C-L and previously in Lanaspá et al¹⁰ and is now provided in Table 1. We respectfully disagree and firmly believe that by using AMPD2^(-/-) mice we can clearly separate the metabolic consequences of F1P accumulation versus AMPD2 activation as aldob/ampd2^(-/-) demonstrate high F1P accumulation with minimal AMPD activity. The use of aldob^(+/-) would be interestingly to identify dose-dependent effects. However, unlike those reports in humans showing that aldob^(+/-) individuals may have some type of susceptibility to hyperuricemia or decreased sensitivity, in mice, no phenotypical or metabolic differences in response to fructose are observed between aldob^(+/+) and aldob^(+/-).

3g) - Line 212: It is not clear from the nomenclature used to describe the mice whether the AMPD2 homozygous are AMPD2^{-/-} (KO homozygous) or +/+ (WT homozygous), although the former is implied since WT mice are used as a control group. It is also not clear whether these are meant to be double-KO mice. Fig2 B does not clarify if these mice are DKOs; DKOs are implied but the text says “AMDP2 heterozygous (KH) and “AMPD2 homozygous (KK)”. The choice of nomenclature for the different strains is not standard.

Response: We fully agree with the reviewer. In this revised version we have aimed to completely standardize the strains limiting the confusion that any reader may have while reading our manuscript. We hope that this revised version provides a more clear nomenclature of the strains employed.

3h) - Lines 216-218: The authors note that DKO mice activate AMPK before exposure to fructose. This points to a pathway that is mitigating fatty liver in the DKO mice that is separate from the accumulation of F1P. AMPD2 deletion is not protecting these mice from damage induced by fructose-feeding; it is protecting them from damage induced by general metabolic imbalance, which then makes fructose-dependent damage less severe.

Response: The reviewer is correct in their appreciation that AMPK is activated in AMPD2 deficient mice. The counter-regulation between AMPK and AMPD2 is not new and we and others have already published on its consequences¹⁴⁻¹⁷. However, at baseline, there is not fatty liver or metabolic imbalance in wild type mice that AMPD2 deletion can protect from. However, aldob^(-/-) mice without fructose exposure tend to have higher liver triglycerides than wild type counterparts. We believe this is because fructose is being endogenously produced from glucose in mice fed a sucrose-free dextrose-rich chow. So the partial metabolic imbalance at baseline in aldob^(-/-) depends also on fructose and fructogenesis and is not a general metabolic imbalance as demonstrated by our published data showing that fructokinase blockade in aldob^(-/-) reduces liver triglycerides at baseline (Lanaspá et al¹⁰, Table 1). Therefore, we believe that any protection observed at baseline in aldob/ampd2^(-/-) mice is related with the consequences associated with endogenous fructose metabolism and not necessarily to general metabolic imbalance.

3i) - Line 243: The authors provide evidence that the DKO mice consume more dietary fructose than aldob^{-/-} mice and state that this “indicates a marked improvement in the tolerance of the (DKO) mice to fructose”. However, in the section starting on line 268, fructose gavage induces severe hypoglycemia in both aldob^{-/-} and DKO mice. It is an over-extrapolation to say that increased feeding in the DKO mice improves fructose tolerance since fructose is not tolerated better by DKO mice in all circumstances. The concentration-dependent, and possible time-dependent, influences of when fructose is better tolerated are important to note.

Response: We agree with the reviewer in that the term “marked” may be an over-extrapolation of the result and as such we have toned down the sentence. We used the term “marked” in the text as the intake of fructose of aldob/ampd2(-/-) mice is significantly higher than aldob(-/-) counterparts, which is very remarkable if we consider that people with HFI have total aversion to fructose. However, and as the reviewer is suggesting, it is correct that the intake of fructose of aldob/ampd2(-/-) mice is substantially lower than the intake of fructose in wild type mice reflecting that aldob/ampd2(-/-) still have issues with this sugar. These issues are probably due to the potential risk of hypoglycemia. We fully agree with the reviewer that the concentration- and time-dependent nature of the study allows aldob/ampd2(-/-) to consume more fructose. As explained above that the acute nature of the oral fructose challenge impairs the proper glucagon response to rescue the hypoglycemic effect. In voluntary drinking however, exposure to a 5% solution, which is 6-10 fold lower dose than the oral fructose challenge, may not induce such a severe hypoglycemic response allowing sufficient time for hypoglycemic counter-acting mechanisms (ex. Glucagon).

3j) - Line 275 claims that AMPD2 deletion blocks the degradation of purines and does not change the accumulation of F1P in HFI patients. Given that there are some physiological differences between mice and humans, the results in aldob-/- mice need to be confirmed in human HFI patients before this statement can be made.

Response: We apologize for the confusion but the sentence on line 275 referred exclusively to the mouse data in figure 4A and its intent is not to suggest that similar findings can be replicated in humans with HFI. We have properly edited the sentence to clarify the statement.

3k) - Line 290 mentions nutrients are lost in the urine but does not state which nutrients. Fig3G and 3H show the excretion of phosphate and glucose; this should be made explicit in the text. The authors should also address why phosphate might be lost in the urine if it is being sequestered inside hepatocytes due to getting trapped on F1P.

Response: In Fanconi syndrome, urinary loss of nutrients include glucose, amino acids and metabolites like phosphate or uric acid. Therefore, measuring fractional excretion of uric acid and phosphate as well as glucosuria is a good standard approach to determine the presence of Fanconi syndrome. Line 290 has been updated with the metabolites commonly found in urine of individuals with Fanconi Syndrome. Also, as suggested, we have clarify the difference between intracellular phosphate sequestration and urinary loss of phosphate. These are two different events with related implications in the overall phenotype. Intrahepatic phosphate levels are low in aldob(-/-) mice. These levels could be improved by taking up phosphate from the circulation. However, the high fractional excretion of phosphate in the urine results in hypophosphatemia¹⁸ which further exacerbates the ability of the liver to take up systemic phosphate.

3l) - The schematic presented in Fig1A implies that F1P directly interacts with AMPD2. There is no evidence that this interaction is direct. This schematic also shows ATP being converted to AMP by KHK when fructose is phosphorylated to F1P, and this is not accurate. ATP is converted to ADP by KHK; conversion of ADP to AMP happens in other reactions.

Response: Corrected.

3m) - In the text, the authors claim that uric acid is ultimately responsible for some of the negative physiological consequences of fructose metabolism in aldob-/- mice. In Fig 3K, aldob-/- mice exposed to either dietary fructose or dietary OxAc have similar amounts of uric acid. However, in Fig 3L, aldob-/- mice that are fed fructose alone, or fed fructose with OxAc, have similar decreases in body weight, while aldob-/- mice fed OxAc alone, but not fructose, do not experience the same decrease in body weight as those fed fructose. This points to a mechanism distinct from uric acid production or accumulation that drives pathology in aldob-/- mice.

Response: We apologize if the reviewed understood from our claim that the accumulation of uric acid is the only culprit in the pathogenesis associated with fructose metabolism in aldob(-/-) mice. Our intent in Figure 3 was to determine whether uric acid is a contributing factor to the failure to thrive observed in fructose-exposed aldob(-/-) mice. In this regard, Figure 3K and 3L demonstrate that OxAc feeding significantly increases uric acid in control and fructose-fed aldob(-/-) mice. When comparing between fructose fed mice, mice receiving fructose

and OxAc demonstrated significant weight loss than mice fed fructose suggesting that the accumulation of uric acid substantially contributes to the deleterious effects of fructose. This is consistent with reports from us and many others demonstrating a key role to uric acid in the pathogenesis of fructose-induced metabolic syndrome¹⁹⁻²⁴. We have edited the paragraph to better clarify our findings and do not overstate the results.

4. There are important controls missing that are needed to support the conclusions drawn in this manuscript. 4a) - If F1P is the driving factor that increases AMPD2 activity and the purine degradation pathway, then aldob^{-/-} mice that are not fed fructose need to be shown as a control for the liver pathology scores and improvement seen in the DKO mice. If aldob^{-/-} mice on a fructose-free diet benefit to the same degree, then F1P is not responsible for activating AMPD2 activity.

Response: As explained above and detailed in new Table1, there are already baseline differences in the liver of aldob^(-/-) on a fructose-free diet. These differences at baseline are not observed when fructokinase is deleted¹⁰ suggesting that they depend on fructose metabolism, F1P accumulation and presumably AMPD2 activation at baseline. Consistently, F1P levels were found to be significantly higher in fructose-free AldoB^(-/-) mice compared to wild type counterparts¹⁰. These F1P levels are now found to be similar to those of wild type mice receiving an acute bolus of fructose (Figure 5A). Further, AMPD activity is also higher in fructose-free AldoB^(-/-) mice compared to wild type counterparts (Figure 1D) without fructose and AMPD2 deletion restores energy charge at baseline (Figure 2E) without modifying baseline F1P levels (Figure 5A). Together, we believe we are providing compelling evidence demonstrating that at both baseline and in fructose-fed aldob^(-/-) mice, the metabolic dysregulation in relies on fructose metabolism and the activation of AMPD2 and the purine degradation pathway.

4b)- If AMPD2 deletion activates GS and allows glycogen to be mobilized to restore energy balance after fructose exposure, then glycogen levels in the DKO mice need to be measured before fructose exposure.

Response: As suggested, we have included in the text glycogen levels in wild type, aldob^(-/-) and aldob/ampd2^(-/-) mice not exposed to dietary fructose. Data is also included in new Table 1. Levels tend to be higher in aldob^(-/-) mice compared with the other groups and exacerbated upon fructose exposure. As already explained in the response, we believe that higher baseline glycogen (and triglycerides) in aldob^(-/-) mice is the result of endogenous fructose production and metabolism from dextrose rich diets as F1P levels are also elevated at baseline and higher triglyceride content at baseline is prevented when fructokinase expression is deleted¹⁰.

4c)- There are no control images shown to support the claim that fructose induces kidney damage in aldob^{-/-} mice. It may be that aldob^{-/-} mice not fed fructose have similar kidney damage.

Response: There is no substantial pathology in Fanconi syndrome as it is rather a metabolic event affecting renal reabsorption of nutrients and electrolytes. In any case, for comparison purposes and as suggested we have included a representative image of the renal cortex of a fructose-free aldob^(-/-) mouse.

4d) While the results presented here are striking, the conclusions drawn are not fully supported by those results. Limitations of this study and alternative hypotheses need to be addressed in the manuscript.

Response: We thank the reviewer for their thorough review and for finding the results striking. We hope that we have been able to address all the concerns and further improve the quality of the manuscript.

Reviewer #3 (Remarks to the Author):

In the current study, the authors Andres-Hernando et al, investigated the metabolic disorders caused by inefficient aldolase B. The study performed in mice that were bred with pure C57BL/6J to obtain both knockout and WT littermates and further crossed with aldolase B KO mice to obtain aldolase b/AMPD2 double KO mice. The mice allowed ad libitum access to 1 % fructose or 2.5 % oxonic acid chow (w/w), compared to fructose-free diet. The authors used histopathology, western blotting, AMPD activity and tissue analysis of various metabolites in order to identify the metabolic pathways activated in hereditary fructose intolerance associated with the progression of the disease.

Results described by the authors including:

- (i) The accumulation of fructose-1-phosphate causes a side-chain reaction including, the purine degradation pathway and a metabolic route initiated by AMPD2 that regulates energy balance.
- (ii) Low amounts of fructose are sufficient to activate AMPD2 in HFI via a phosphate trap
- (iii) Loss of AMPD2 in aldolase B deficient mice, corrects metabolic malfunction and amends nonalcoholic steatohepatitis and liver disease while increasing the tolerance for fructose.

The authors concluded that this manuscript reveals new important metabolic pathways triggered by fructose metabolism that can be targeted to improve metabolic dysfunction in HFI.

A major concern is the lack of depth and granularity in the experimental design and presentation of the data. The following are highly recommended:

1. Regarding experimental design, it is very important to identify the function and dysfunction of a specific metabolic pathway by using stable isotope tracer, such as ¹³C-labeled fructose given by oral gavage or IV infusion. The authors can trace the exact metabolism of fructose determine specific activity, such as flux rate and determining the site of malfunction. Examples of such investigation can be seen in: Biochem J. 2012 Jun 15;444(3):537-51; Am J Sports Med 2018 Jul;46(9):2222-2231; Cell Metab. 2020 Jan 7; 31(1): 174–188.e7.

Response: We have previously shown that similar to phosphate, fructose is also trapped as F1P in aldob(-/-) mice dose- and time dependently¹⁰. This suggests that labeling fructose to trace its metabolism would not provide more information than the already presented showing that it is accumulated as F1P unless it is also metabolized by collateral pathways that may contribute to the disease. However, and as we published¹⁰, blocking fructokinase fully prevents the deleterious effects that fructose induce in aldob(-/-) mice indicating that this is the main pathway engaged in the pathogenesis of fructose induced liver disease in aldob(-/-) mice. This observation also suggests that both the accumulation of F1P and/or the activation of the AMPD2 side chain pathway are the only potential downstream consequences and thus mechanisms associated with the disease.

2. To specifically determine the liver metabolic malfunction in WT versus aldolase B KO and aldolase b/AMPD2 double knockout, it is prefer using liver perfusion experiment with ¹³C-labeled fructose or another precursor. The in vivo study as presented in the current paper cannot specifically differentiate between metabolism in the liver and the other organs in the body.

Response: We thank the reviewer for their suggestion. In 2021, we published our first work with AMPD2 tissue-specific knockouts including hepatocyte-specific AMPD2 deficient mice²⁵. As suggested by the reviewer, in order to fully ascertain and parse out the importance of AMPD2 activation in liver metabolic malfunction induced by fructose, we have conducted experiments in which AMPD2 is specifically deleted in hepatocytes of aldob(-/-) mice. Remarkably, the protective effects observed against fructose in whole body AMPD2(-/-) mice are replicated in aldob(-/-) liver-specific deficient for AMPD2 thus emphasizing the importance that hepatic fructose metabolism has in the overall phenotype observed in aldob(-/-) mice. Further, and as mentioned above in point 2c of reviewer's 2 response and outside the scope of this paper, we have also conducted pilot data to determine tissue-specific effects of fructokinase blockade in aldob(-/-) mice observing that the deleterious of fructokinase in the liver is sufficient to prevent F1P accumulation and the deleterious consequences of fructose -including acute hypoglycemia- while its blockade in extrahepatic organs like the gut exacerbate the phenotype probably due to higher delivery of fructose to the liver consistent with we and others previously demonstrated^{9,12}. A new figure with liver-specific AMPD2(-/-) mice -named aldob/ampd2^{F1/F1}xAlbCre(-/-)- has been added to the manuscript.

3. It is highly recommended that the authors provide a companion figure, e.g., heatmap to show the markers used to defining the metabolic disorder and to help navigate the identity of a specific metabolic site and enable comparison to the overall metabolic function (data in Fig 1 D-G, Fig 2 C-G, M, N, Fig 3, A-H).

Response: We have not run metabolomics or multi-omics analysis in our samples as this is a hypothesis-driven study aimed to determine the relative importance and separate the downstream consequences of F1P accumulation and AMPD2 activation in fructose-exposed aldob(-/-) mice. However, we agree with the reviewer in the need of having a figure-type element in which the parameters analyzed to establish metabolic dysfunction can be comparable between all the strains employed. Therefore, now we provide a new table (Table1) with all the parameters measured in plasma and liver at baseline and after 2-week fructose (1% w/w) study for all strains including wild type, aldob(-/-), aldob/ampd2(-/-) and aldob/ampd2^{F1/F1}-AlbCre(-/-) mice.

References:

- 1 Pilar Lopez, M., Gomez-Lechon, M. J. & Castell, J. V. Role of glucose, insulin, and glucagon in glycogen mobilization in human hepatocytes. *Diabetes* **40**, 263-268, doi:10.2337/diabetes.40.2.263 (1991).
- 2 Aldámiz-Echevarría, L. *et al.* Non-alcoholic fatty liver in hereditary fructose intolerance. *Clinical Nutrition* **39**, 455-459, doi:10.1016/j.clnu.2019.02.019 (2020).
- 3 Morris, R. C., Jr. An experimental renal acidification defect in patients with hereditary fructose intolerance. II. Its distinction from classic renal tubular acidosis; its resemblance to the renal acidification defect associated with the Fanconi syndrome of children with cystinosis. *J Clin Invest* **47**, 1648-1663, doi:10.1172/JCI105856 (1968).
- 4 Simons, N. *et al.* Kidney and vascular function in adult patients with hereditary fructose intolerance. *Mol Genet Metab Rep* **23**, 100600, doi:10.1016/j.ymgmr.2020.100600 (2020).
- 5 James, C. L., Rellos, P., Ali, M., Heeley, A. F. & Cox, T. M. Neonatal screening for hereditary fructose intolerance: frequency of the most common mutant aldolase B allele (A149P) in the British population. *J Med Genet* **33**, 837-841, doi:10.1136/jmg.33.10.837 (1996).
- 6 Coffee, E. M., Yerkes, L., Ewen, E. P., Zee, T. & Tolan, D. R. Increased prevalence of mutant null alleles that cause hereditary fructose intolerance in the American population. *J Inherit Metab Dis* **33**, 33-42, doi:10.1007/s10545-009-9008-7 (2010).
- 7 Softic, S., Cohen, D. E. & Kahn, C. R. Role of Dietary Fructose and Hepatic De Novo Lipogenesis in Fatty Liver Disease. *Dig Dis Sci* **61**, 1282-1293, doi:10.1007/s10620-016-4054-0 (2016).
- 8 Zhao, S. *et al.* Dietary fructose feeds hepatic lipogenesis via microbiota-derived acetate. *Nature* **579**, 586-591, doi:10.1038/s41586-020-2101-7 (2020).
- 9 Andres-Hernando, A. *et al.* Deletion of Fructokinase in the Liver or in the Intestine Reveals Differential Effects on Sugar-Induced Metabolic Dysfunction. *Cell Metab* **32**, 117-127 e113, doi:10.1016/j.cmet.2020.05.012 (2020).
- 10 Lanaspá, M. A. *et al.* Ketohexokinase C blockade ameliorates fructose-induced metabolic dysfunction in fructose-sensitive mice. *J Clin Invest* **128**, 2226-2238, doi:10.1172/JCI94427 (2018).
- 11 Andres-Hernando, A. *et al.* Endogenous Fructose Production and Metabolism Drive Metabolic Dysregulation and Liver Disease in Mice with Hereditary Fructose Intolerance. *Nutrients* **15**, doi:10.3390/nu15204376 (2023).
- 12 Jang, C. *et al.* The small intestine shields the liver from fructose-induced steatosis. *Nat Metab* **2**, 586-593, doi:10.1038/s42255-020-0222-9 (2020).
- 13 Park, S. H. *et al.* Fructose induced KHK-C can increase ER stress independent of its effect on lipogenesis to drive liver disease in diet-induced and genetic models of NAFLD. *Metabolism* **145**, 155591, doi:10.1016/j.metabol.2023.155591 (2023).
- 14 Zabielska, M. A., Borkowski, T., Slominska, E. M. & Smolenski, R. T. Inhibition of AMP deaminase as therapeutic target in cardiovascular pathology. *Pharmacol Rep* **67**, 682-688, doi:10.1016/j.pharep.2015.04.007 (2015).
- 15 Lanaspá, M. A. *et al.* Opposing activity changes in AMP deaminase and AMP-activated protein kinase in the hibernating ground squirrel. *PLoS One* **10**, e0123509, doi:10.1371/journal.pone.0123509 (2015).

- 16 Plaideau, C. *et al.* Effects of pharmacological AMP deaminase inhibition and *Ampd1* deletion on nucleotide levels and AMPK activation in contracting skeletal muscle. *Chem Biol* **21**, 1497-1510, doi:10.1016/j.chembiol.2014.09.013 (2014).
- 17 Lanaspa, M. A. *et al.* Counteracting roles of AMP deaminase and AMP kinase in the development of fatty liver. *PLoS One* **7**, e48801, doi:10.1371/journal.pone.0048801 (2012).
- 18 Karatzas, A. *et al.* Fanconi syndrome in the adulthood. The role of early diagnosis and treatment. *J Musculoskelet Neuronal Interact* **17**, 303-306 (2017).
- 19 Russo, E. *et al.* Fructose and Uric Acid: Major Mediators of Cardiovascular Disease Risk Starting at Pediatric Age. *Int J Mol Sci* **21**, doi:10.3390/ijms21124479 (2020).
- 20 Chen, G. & Jia, P. Allopurinol decreases serum uric acid level and intestinal glucose transporter-5 expression in rats with fructose-induced hyperuricemia. *Pharmacol Rep* **68**, 782-786, doi:10.1016/j.pharep.2016.04.014 (2016).
- 21 Asghar, Z. A. *et al.* Maternal fructose drives placental uric acid production leading to adverse fetal outcomes. *Sci Rep* **6**, 25091, doi:10.1038/srep25091 (2016).
- 22 Lanaspa, M. A. *et al.* Uric acid induces hepatic steatosis by generation of mitochondrial oxidative stress: potential role in fructose-dependent and -independent fatty liver. *J Biol Chem* **287**, 40732-40744, doi:10.1074/jbc.M112.399899 (2012).
- 23 Perez-Pozo, S. E. *et al.* Excessive fructose intake induces the features of metabolic syndrome in healthy adult men: role of uric acid in the hypertensive response. *Int J Obes (Lond)* **34**, 454-461, doi:10.1038/ijo.2009.259 (2010).
- 24 Nakagawa, T. *et al.* A causal role for uric acid in fructose-induced metabolic syndrome. *Am J Physiol Renal Physiol* **290**, F625-631, doi:10.1152/ajprenal.00140.2005 (2006).
- 25 Andres-Hernando, A. *et al.* Umami-induced obesity and metabolic syndrome is mediated by nucleotide degradation and uric acid generation. *Nat Metab* **3**, 1189-1201, doi:10.1038/s42255-021-00454-z (2021).

Reviewers' comments:

Reviewer #1 (Remarks to the Author):

"Activation of side-chain AMPD2 drives metabolic dysregulation and liver disease in mice with hereditary fructose intolerance."

The manuscript is properly revised according to the comments of reviewers.

Reviewer #2 (Remarks to the Author):

Review of COMMSBIO-23-2409A: Activation of side-chain AMPD2 drives metabolic dysregulation and liver disease in mice with hereditary fructose intolerance.

Response to rebuttal letter:

The authors state that the sucrose-free diet is very rich in complex carbohydrates (50% dextrose and 12.9% corn starch). Dextrose is the same thing as D-glucose and it is not a complex carbohydrate. The authors make the claim that this level of glucose is activating the polyol pathway to endogenously produce fructose, but do not suggest any alternative hypotheses that could also contribute to energy charge or changes in AMP/ATP levels that could also be present with aldolase B deficiency.

General comments:

There are still a lot of typographical and grammatical errors reduce readability. The progression between important ideas is not always delineated or logical; the manuscript would benefit from splitting some of the larger paragraphs into smaller paragraphs that each discuss one idea. Since multiple diets and time points for feeding were used, the figures would benefit from a schematic that shows experimental design of when different groups of mice went on different diets and for how long.

The current title, and much of the language in the manuscript, suggest that the contribution of AMPD2 activity and subsequent production of uric acid are the main drivers of metabolic dysregulation and liver disease in *aldoB*^{-/-} mice. While AMPD2 deficiency does ameliorate some aspects of metabolic dysregulation, it is not clear that this is the main driver of liver disease. A phosphate-sink created by F1P cannot explain all of the metabolic phenomenon reported here, since *aldoB*^{-/-} mice fed a sucrose-free diet have similarly low [P] as WT mice fed a fructose diet, but sucrose-free fed *aldoB*^{-/-} mice do not have increased AMPD activity compared to fructose-fed WT mice. The authors present evidence that supports the conclusion that dietary fructose in *aldoB*^{-/-} mice does activate the AMPD pathway and claim that AMPD activation is dependent on F1P levels. The authors do not offer an explanation for why elevated F1P in sucrose-free fed *aldoB*^{-/-} mice do not activate the pathway that is allegedly responsible for the metabolic dysregulation seen in hepatocytes, even though sucrose-free fed *aldoB*^{-/-} mice have elevated TG, increased AST and ALT levels, and increased liver and kidney inflammation. Alternative, or complementary, explanations for the metabolic phenotypes seen in *aldoB*^{-/-} mice should be acknowledged and discussed.

Specific comments:

The methods don't mention how liver triglycerides (TG) were measured. The results report the liver TG content as 1.2 +/- 0.3 g/g. This implies that there is more than 100% of the liver g is made of TGs.

The authors state that *aldoB*^{-/-} mice fed a sucrose-free diet have increased [fru-1P], which acts as a phosphate sink that lowers intracellular [P] and activates AMPD2. The authors also state that AMPD expression was lower (though not significantly; Fig 1B) and AMPD activity was higher (Fig 1D) in *aldoB*^{-/-} than WT mice, when *aldoB*^{-/-} mice are fed a sucrose-free diet. Is there an explanation for how AMPD expression could be lower, but activity is higher?

In lines 234-244, the authors report that *aldoB*^{-/-} mice on a sucrose-free diet have elevated liver TG, that exposure to 1% fructose (w/w) resulted in a marked exacerbation of the metabolic phenotype (Fig2H-O), and that the double knock-out *aldoB/ampD2*^{-/-} mice have reduced liver lipid accumulation (Fig2J-K). Table 1 also shows that *aldoB*^{-/-} mice exposed to 1% fructose for 5 wks have a reduced liver lipid accumulation, while the *aldoB*^{-/-} mice fed a sucrose-free diet do not have increased lipid accumulation compared to WT mice. These discrepancies need further explanation.

If AMPD2 deletion restores energy charge at baseline (Fig. 2E) without modifying fru-1P levels (Fig. 5A), then how is the metabolic dysregulation dependent on fructose metabolism? Doesn't this point to the regulation of phosphate in general and not the catabolism of fru-1P specifically?

The authors state that AMPD2 deletion results in a baseline dose-dependent effect in *aldoB*^{-/-} mice for metabolic markers including hepatic energy charge and AMP/ATP ratio (Fig2E,F). It is unclear how a metric can be at baseline, and also be subject to different doses of a reagent/stimulus. A more detailed description of what is meant by baseline and dose-dependent in this context is needed.

The authors quantify the aversion of *aldoB*^{-/-} mice to dietary fructose by measuring food intake and volume of sweetened beverages consumed. In the text, authors note that deletion of AMPD2 results in an increase in sweetened beverage intake of the double knock-out *aldoB/ampD2*^{-/-} mice compared to *aldoB*^{-/-} mice, but they do not state that the beverage intake is still less than that of WT mice, and that the decreased sweetened beverage intake may be significantly different for *aldoB/ampD2*^{-/-} mice exposed to fructose (2.5% (w/v)) and HFCS (5% (w/v)) compared to WT. The authors do not comment on what looks like a dose-dependent trend in avoiding fructose-sweetened water between the 2.5% and 5% solutions. Does this trend correlate with the concentration of F1P and/or uric acid levels in the double knock-out mice? The hypothesis that F1P increases can be mitigated by reducing uric acid production via the blockade of AMPD2 deficiency does not seem to apply in this scenario.

The authors postulate that one reason HFI patients and *aldoB*^{-/-} mice show a failure to thrive is due to reduced kidney function seen in Falconi Syndrome. They speculate that the increased uric acid in *aldoB*^{-/-} patients and mice lead to kidney damage (Fig3I) and that further increases in uric acid would worsen this kidney damage. The authors show that feeding *aldoB*^{-/-} mice 2.5% (w/w) OxAC increases uric acid and weight loss (Fig3K,L) but the histology for kidneys after OxAC treatment is missing.

In the text of the results, weight loss of *aldoB*^{-/-} mice fed a 1% fructose diet is reported (Fig3C). The percent change in body weight after 8wks is shown for *aldoB*^{-/-} mice fed either a sucrose-free diet, a 1% (w/w) fructose diet, or for *aldoB/ampD2*^{-/-} mice fed a sucrose-free diet. If loss of AMPD2 activity reduces metabolic phenotypes in an *aldoB*^{-/-} background, why is the percent change in body weight for the *aldoB/ampD2*^{-/-} mice fed a 1% fructose (w/w) diet not shown?

For *aldoB*^{-/-} mice fructose-fed a 1% fructose + 2.5% (w/w) OxAc is described as a 57% reduction in body weight gain compared to vehicle-treated *aldoB*^{-/-} mice (Fig3L) This difference in describing the loss of body weight is confusing. The importance of the contribution of OxAc would be better described by comparing the weight differences between *aldoB*^{-/-} mice on a sucrose diet, with or without 2.5% (w/w) OxAc, vs. *aldoB*^{-/-} mice on a 1% fructose diet, with or without 2.5% (w/w) OxAc. It isn't clear if OxAc exacerbates the effects of fructose-feeding, as the text suggests.

Reviewer #3 (Remarks to the Author):

As indicated before, the current manuscript represents important data regarding the relationship between hereditary fructose intolerance (HFI), the hepatic fructose consumption, the AMPD2, and thus, the NAFLD.

Overall, the authors addressed most of the reviewers 1 and 2 concern, and also my (Review #3), concern. However, the authors response to my suggestion for performing ¹³C tracer metabolomics analysis was : “that labeling fructose to trace its metabolism would not provide more information than the already presented”, and “We have not run metabolomics or multi-omics analysis in our samples as this is a hypothesis-driven study”. In this regard, I completely disagree with the authors. The tracer and metabolomics analysis are essential for determining the site of function and malfunction of any metabolic pathway and may add an important information regarding the HFI process which could be used to reduce the HFI disease.

Reviewer #1 (Remarks to the Author):

"Activation of side-chain AMPD2 drives metabolic dysregulation and liver disease in mice with hereditary fructose intolerance."

The manuscript is properly revised according to the comments of reviewers.

Response: *Thank you very much for considering we have revised the manuscript properly.*

Reviewer #2 (Remarks to the Author):

Review of COMMSBIO-23-2409A: Activation of side-chain AMPD2 drives metabolic dysregulation and liver disease in mice with hereditary fructose intolerance.

Response to rebuttal letter:

The authors state that the sucrose-free diet is very rich in complex carbohydrates (50% dextrose and 12.9% corn starch). Dextrose is the same thing as D-glucose and it is not a complex carbohydrate. The authors make the claim that this level of glucose is activating the polyol pathway to endogenously produce fructose, but do not suggest any alternative hypotheses that could also contribute to energy charge or changes in AMP/ATP levels that could also be present with aldolase B deficiency.

General comments:

There are still a lot of typographical and grammatical errors reduce readability. The progression between important ideas is not always delineated or logical; the manuscript would benefit from splitting some of the larger paragraphs into smaller paragraphs that each discuss one idea. Since multiple diets and time points for feeding were used, the figures would benefit from a schematic that shows experimental design of when different groups of mice went on different diets and for how long.

Response: We agree with the reviewer. As suggested we have reduced the length of the paragraphs splitting them into shorter texts. Similarly, we have added a schematic showing the experimental design of each study to figure 3 which is the one that includes different studies. The oxonic acid data in previous Figure 3 has been moved to new Figure 4 and the text expanded (lines 280-290).

*The current title, and much of the language in the manuscript, suggest that the contribution of AMPD2 activity and subsequent production of uric acid are the main drivers of metabolic dysregulation and liver disease in *aldoB*^{-/-} mice. While AMPD2 deficiency does ameliorate some aspects of metabolic dysregulation, it is not clear that this is the main driver of liver disease. A phosphate-sink created by F1P cannot explain all of the metabolic phenomenon reported here, since *aldoB*^{-/-} mice fed a sucrose-free diet have similarly low [P] as WT mice fed a fructose diet, but sucrose-free fed *aldoB*^{-/-} mice do not have increased AMPD activity compared to fructose-fed WT mice. The authors present evidence that supports the conclusion that dietary fructose in *aldoB*^{-/-} mice does activate the AMPD pathway and claim that AMPD activation is dependent on F1P levels. The authors do not offer an explanation for why elevated F1P in sucrose-free fed *aldoB*^{-/-} mice do not activate the pathway that is allegedly responsible for the metabolic dysregulation seen in hepatocytes, even though sucrose-free fed *aldoB*^{-/-} mice have elevated TG, increased AST and ALT levels, and increased liver and kidney inflammation. Alternative, or complementary, explanations for the metabolic phenotypes seen in *aldoB*^{-/-} mice should be acknowledged and discussed.*

Response: When comparing wild type (*aldob*^{+/+}) versus *aldob*^{-/-} mice on sucrose-free diet, *aldoB*^{-/-} demonstrate lower [P] (69.93±1.73 vs 39.42±6.55 nmol/mg P<0.01, Fig. 1B), higher AMPD activity (8.4±2.2 vs 17.9±3.1 umol ammonia/min/mg, P<0.01, Fig. 1D) and lower AMP (3.58±0.8 vs 1.4±0.5 nmol/mg, P<0.01 Fig. 1E) indicating that the pathway is activated at baseline in sucrose-free *aldob*^{-/-} mice. We propose that this explains the higher expression of markers of liver dysfunction and injury in *aldob*^{-/-} mice at baseline. We have modified the first paragraph of the results section to better clarify that there are already baseline significant differences in phosphate and AMPD activation between sucrose-free fed wild type and *aldob*^{-/-} mice. The modified paragraph:

*"AMPD2 expression is not significantly different between wild type and *aldob*^{-/-} mice (Fig. 1B). However, intracellular phosphate levels are significantly lower both at baseline (69.93±1.73 vs 39.42±6.55 nmol/mg P<0.01) and upon fructose*

exposure (47.27 ± 5.71 vs 19.82 ± 7.31 nmol/mg $P < 0.01$, Fig 1C) in *aldob(-/-)* mice compared to wild type counterparts. In turn, AMPD activity was found to be significantly higher in *aldob(-/-)* mice compared to wild type counterparts at baseline (8.4 ± 2.2 vs 17.9 ± 3.1 umol ammonia/min/mg, $P < 0.01$) and further exacerbated in mice fed a diet containing fructose (12.6 ± 2.6 vs 41.8 ± 7.5 umol ammonia/min/mg, $P < 0.01$, Fig. 1D) for 2 weeks (1 % w/w).” **Lines 196-203**

We propose that baseline differences could be due to endogenous production and metabolism of fructose similar to what we proposed previously¹. We appreciate the reviewer’s comment acknowledging that AMPD2 deletion ameliorates metabolic dysregulation in *aldob(-/-)*. We do not imply that the activation of AMPD2 is an exclusive mechanism or the main single driver of the disease and neither the title indicates that although we believe that our data supports the concept that activation of AMPD2 is an important deleterious step in the pathogenesis of the disease. We have added a paragraph in the discussion section acknowledging that other metabolic routes besides the purine degradation pathway may be dysfunctional in HFI (**lines 398-404**) and propose (as suggested by reviewer 3) that metabolomic or tracer studies could provide a much clearer picture of how inefficient fructose metabolism affects energy state and overall metabolism in *aldob(-/-)* mice.

Specific comments:

The methods don’t mention how liver triglycerides (TG) were measured. The results report the liver TG content as 1.2 +/- 0.3 g/g. This implies that there is more than 100% of the liver g is made of TGs.

Response: TG determination was carried out using a commercially available kit (ETGA-200, Bioassay Systems) now included in the methods section (**lines 171-173**). The TG data obtained is normalized by soluble protein in the lysate and not by whole tissue weight.

The authors state that *aldoB(-/-)* mice fed a sucrose-free diet have increased [fru-1P], which acts as a phosphate sink that lowers intracellular [P] and activates AMPD2. The authors also state that AMPD expression was lower (though not significantly; Fig 1B) and AMPD activity was higher (Fig 1D) in *aldoB(-/-)* than WT mice, when *aldoB(-/-)* mice are fed a sucrose-free diet. Is there an explanation for how AMPD expression could be lower, but activity is higher?

Response: Phosphate regulates AMPD activity and not necessarily the expression of any of its isoforms. Expression of AMPD2 is not significantly different between wild type and *aldob(-/-)* mice. A tendency towards lower expression would suggest to us a compensatory mechanism in response to the hyperactivation. This negative feedback is shown in a variety of biological processes, for example, uric acid inhibits activity of xanthine oxidase, the enzyme that produces uric acid in the purine degradation pathway while allopurinol up-regulates xanthine oxidase expression.

In lines 234-244, the authors report that *aldoB(-/-)* mice on a sucrose-free diet have elevated liver TG, that exposure to 1% fructose (w/w) resulted in a marked exacerbation of the metabolic phenotype (Fig2H-O), and that the double knock-out *aldoB/ampd2(-/-)* mice have reduced liver lipid accumulation (Fig2J-K). Table 1 also shows that *aldoB(-/-)* mice exposed to 1% fructose for 5 wks have a reduced liver lipid accumulation, while the *aldoB(-/-)* mice fed a sucrose-free diet do not have increased lipid accumulation compared to WT mice. These discrepancies need further explanation.

Response: We thank the reviewer for the careful review and pointing out to this discrepancy. The data in table 1 was pooled from both the first studies (comparing wild type vs *aldob* vs AMPD2 whole body KO) and the following study after the first submission (comparing *aldob(-/-)* vs liver specific AMPD2 KO) but it was not normalized and it should as they were performed at different times. We have tried now carefully analyze the data and have corrected the table accordingly.

If AMPD2 deletion restores energy charge at baseline (Fig. 2E) without modifying fru-1P levels (Fig. 5A), then how is the metabolic dysregulation dependent on fructose metabolism? Doesn’t this point to the regulation of phosphate in general and not the catabolism of fru-1P specifically?

Response: Energy charge is different at baseline between wild type and *aldob(-/-)* mice (0.93 ± 0.01 vs 0.89 ± 0.01 , $P < 0.01$) along with lower baseline phosphate (69.93 ± 1.73 vs 39.42 ± 6.55 nmol/mg $P < 0.01$) and higher AMPD activity (8.4 ± 2.2 vs 17.9 ± 3.1 umol ammonia/min/mg, $P < 0.01$). F1P is also significantly higher in *aldob(-/-)* compared to wild type mice at baseline (15.5 ± 4.0 vs 112.8 ± 21.2 AU, $P < 0.01$). This suggests that fructose is already being metabolized in mice that are in a fructose-free diet resulting in both F1P accumulation and AMPD activation in *aldob(-/-)* possibly as a consequence of endogenous fructose production which we have shown takes place in *aldob(-/-)* mice¹. However, this effect is exacerbated upon dietary fructose exposure. The restoration of energy charge without a reduction in F1P suggests to us that the activation of AMPD mediates

energy charge and not F1P. We agree with the reviewer that this could potentially point to a more specific effect of phosphate in overall energy charge in other biological processes besides fructose metabolism. Consistently, we have shown that the activation of AMPD2 independently of fructose also drives metabolic dysregulation like in the case of umami-rich foods².

The authors state that AMPD2 deletion results in a baseline dose-dependent effect in *aldoB*^{-/-} mice for metabolic markers including hepatic energy charge and AMP/ATP ratio (Fig2E,F). It is unclear how a metric can be at baseline, and also be subject to different doses of a reagent/stimulus. A more detailed description of what is meant by baseline and dose-dependent in this context is needed.

Response: We apologize if the statement is unclear. We have documented an intermediate effect in energy charge and AMP/ATP ratio between *aldob*(^{-/-}) mice expressing 100% AMPD2, 50% AMPD2 (heterozygous) or 0% AMPD2 (knockouts). This range is what we denominated dose-dependent. We agree with the reviewer as the term seems vague and it does not clearly define the study. Therefore, we have corrected the text in order to better clarify what we meant by dose-dependently (lines 226-228).

The authors quantify the aversion of *aldoB*^{-/-} mice to dietary fructose by measuring food intake and volume of sweetened beverages consumed. In the text, authors note that deletion of AMPD2 results in an increase in sweetened beverage intake of the double knock-out *aldoB/ampD2*^{-/-} mice compared to *aldoB*^{-/-} mice, but they do not state that the beverage intake is still less than that of WT mice, and that the decreased sweetened beverage intake may be significantly different for *aldoB/ampD2*^{-/-} mice exposed to fructose (2.5% (w/v)) and HFCS (5% (w/v)) compared to WT. The authors do not comment on what looks like a dose-dependent trend in avoiding fructose-sweetened water between the 2.5% and 5% solutions. Does this trend correlate with the concentration of F1P and/or uric acid levels in the double knock-out mice? The hypothesis that F1P increases can be mitigated by reducing uric acid production via the blockade of AMPD2 deficiency does not seem to apply in this scenario.

Response: As indicated, we now state that *aldob/ampd2* (^{-/-}) mice have an intermediate and significantly different intake of fructose between wild type and *aldob*(^{-/-}) mice. Based on our data in Fig 6A (previous Fig 5A) and as the reviewer suggests we think that F1P are not different and still high in *aldob/ampd2* (^{-/-}) mice. To us, this point to effects related with blockade of AMPD2 metabolism extrahepatically. In this regard, we have shown that blocking KHK and fructose metabolism reduces fructose intake and preference³. This was observed in whole body and gut-specific fructokinase KO and not in liver-specific fructokinase KO and suggests the presence of a gut-brain axis mediated by fructose metabolism that controls fructose intake. In confidentiality for the reviewer and outside the focus of this article, our lab has data demonstrating that similar to fructokinase knockout mice, AMPD2 KO mice also have reduced intake and preference for fructose. As a result, in *aldob/ampd2* (^{-/-}) mice there is a conflicting scenario taking place in which on one hand the improved metabolic dysregulation allow mice to consume more fructose but on the other hand there is a reduction in the hedonic pleasure obtained from fructose and sugar that limits the overall intake. Thus, we believe that the consequence of these conflicting events results in the intermediate intake of sugar observed in *aldob/ampd2* (^{-/-}) mice. As suggested by the reviewer, we have added an statement acknowledging the intermediate intake of fructose between wild type and *aldob*(^{-/-}) mice (lines 256-259).

The authors postulate that one reason HFI patients and *aldoB*^{-/-} mice show a failure to thrive is due to reduced kidney function seen in Falconi Syndrome. They speculate that the increased uric acid in *aldoB*^{-/-} patients and mice lead to kidney damage (Fig3I) and that further increases in uric acid would worsen this kidney damage. The authors show that feeding *aldoB*^{-/-} mice 2.5% (w/w) OxAC increases uric acid and weight loss (Fig3K,L) but the histology for kidneys after OxAC treatment is missing.

Response: We have expanded new Figure 4 including histology and urinary profile of *aldob*(^{-/-}) mice after OxAc treatment. As shown in the figure, nutrient waste is very evident in fructose and OxAc treated *aldob*(^{-/-}) mice. Even though Fanconi is more a metabolic event and traditionally associated with limited pathology, uric acid levels in fructose and OxAc treated mice are very high and kidney injury is prominent similar to that found on uricase knockout mice with spontaneous hyperuricemia⁴ (lines 285-290).

In the text of the results, weight loss of *aldoB*^{-/-} mice fed a 1% fructose diet is reported (Fig3C). The percent change in body weight after 8wks is shown for *aldoB*^{-/-} mice fed either a sucrose-free diet, a 1% (w/w) fructose diet, or for *aldoB/ampD2*^{-/-} mice fed a sucrose-free diet. If loss of AMPD2 activity reduces metabolic phenotypes in an *aldoB*^{-/-} background, why is the percent change in body weight for the *aldoB/ampD2*^{-/-} mice fed a 1% fructose (w/w) diet not shown?

Response: The data in Fig. 3C-H correspond to wild type (aldob+/+), aldob(-/-) and aldob/ampd2 (-/-) on a 1% fructose diet and not on suc-free diet. We thank the reviewer for pointing out as they were represented as clear symbols which made the reviewer consider sucrose-free diet according to the legend in Fig. 3B. We have edited the symbols from clear to solid colors to indicate that the data presented in Fig. 3C-H os for fructose-fed animals.

For aldob-/- mice fructose-fed a 1% fructose + 2.5% (w/w) OxAc is described as a 57% reduction in body weight gain compared to vehicle-treated aldob-/- mice (Fig3L) This difference in describing the loss of body weight is confusing. The importance of the contribution of OxAc would be better described by comparing the weight differences between aldob-/- mice on a sucrose diet, with or without 2.5% (w/w) OxAc, vs. aldob-/- mice on a 1% fructose diet, with or without 2.5% (w/w) OxAc. It isn't clear if OxAc exacerbates the effects of fructose-feeding, as the text suggests.

Response: We agree with the reviewer. We have edited the text accordingly to better describe the contribution of OxAc in control and fructose exposed mice (lines 285-286).

Reviewer #3 (Remarks to the Author):

As indicated before, the current manuscript represents important data regarding the relationship between hereditary fructose intolerance (HFI), the hepatic fructose consumption, the AMPD2, and thus, the NAFLD. Overall, the authors addressed most of the reviewers 1 and 2 concern, and also my (Review #3), concern. However, the authors response to my suggestion for performing 13C tracer metabolomics analysis was : "that labeling fructose to trace its metabolism would not provide more information than the already presented", and "We have not run metabolomics or multi-omics analysis in our samples as this is a hypothesis-driven study". In this regard, I completely disagree with the authors. The tracer and metabolomics analysis are essential for determining the site of function and malfunction of any metabolic pathway and may add an important information regarding the HFI process which could be used to reduce the HFI disease.

Response: We thank the reviewer for considering that we provide important data regarding how liver disease is developed in HFI and considering we addressed his/her concerns. Also, we apologize as it was not our intent to minimize the clinical importance of tracing studies and the potential relevant information it could be obtained in our model. We believe that tracing studies would fall outside the scope if this manuscript but we are very interested in conducting studies analyzing the fate of fructose in aldolase b knockout mice. We currently do not have the ability to run these types of studies but we are reaching out to colleagues to run them in the near future. Further, we have added a paragraph to the limitations of the study (lines 398-404) indicating that tracing studies with 13C-fructose would be particularly informative to get a much clear picture of how fructose is metabolized and its consequences in aldolase b deficient mice.

References:

- 1 Andres-Hernando, A. *et al.* Endogenous Fructose Production and Metabolism Drive Metabolic Dysregulation and Liver Disease in Mice with Hereditary Fructose Intolerance. *Nutrients* **15**, doi:10.3390/nu15204376 (2023).
- 2 Andres-Hernando, A. *et al.* Umami-induced obesity and metabolic syndrome is mediated by nucleotide degradation and uric acid generation. *Nat Metab* **3**, 1189-1201, doi:10.1038/s42255-021-00454-z (2021).
- 3 Andres-Hernando, A. *et al.* Deletion of Fructokinase in the Liver or in the Intestine Reveals Differential Effects on Sugar-Induced Metabolic Dysfunction. *Cell Metab* **32**, 117-127 e113, doi:10.1016/j.cmet.2020.05.012 (2020).
- 4 Lu, J. *et al.* Knockout of the urate oxidase gene provides a stable mouse model of hyperuricemia associated with metabolic disorders. *Kidney Int* **93**, 69-80, doi:10.1016/j.kint.2017.04.031 (2018).

Reviewers' comments:

Reviewer #2 (Remarks to the Author):

Review of COMMSBIO-23-2409A: Activation of side-chain AMPD2 drives metabolic dysregulation and liver disease in mice with hereditary fructose intolerance.

Reviewer comments:

The authors have addressed many of the concerns from the last review. However, the manuscript continues to suffer from numerous grammatical errors, ambiguous clauses, and general disorganized writing.

For example, line 195-6, "AMPD2 activation in *aldoB*(-/-) is the consequence of low intracellular phosphate, a known AMPD inhibitor, as it is trapped as F1P". It seems that the authors mean to say that Intracellular phosphate is a known AMPD inhibitor. The sequestration of intracellular phosphate on F1P results in low intracellular levels in *aldoB*(-/-) mice and activation of AMPD. There is still inconsistent use of "*aldoB*(-/-)" and "aldolase B KO". Line 226-8: There seems to be a typo in the genotypes mentioned since AMPD2(+/-) is mentioned twice.

Fig1 panels H, I, and J and Fig2 panel F show Western Blots without the actin control.

Reviewer #2 (Remarks to the Author):

Review of COMMSBIO-23-2409A: Activation of side-chain AMPD2 drives metabolic dysregulation and liver disease in mice with hereditary fructose intolerance.

The authors have addressed many of the concerns from the last review. However, the manuscript continues to suffer from numerous grammatical errors, ambiguous clauses, and general disorganized writing.

For example, line 195-6, "AMPD2 activation in aldoB(-/-) is the consequence of low intracellular phosphate, a known AMPD inhibitor, as it is trapped as F1P". It seems that the authors mean to say that Intracellular phosphate is a known AMPD inhibitor. The sequestration of intracellular phosphate on F1P results in low intracellular levels in aldoB(-/-) mice and activation of AMPD. There is still inconsistent use of "aldoB(-/-)" and "aldolase B KO". Line 226-8: There seems to be a typo in the genotypes mentioned since AMPD2(+/-) is mentioned twice.

Response: *Thank you very much for considering that we have addressed many of the concerns from the last review. For this revised version, we have performed a thorough review of the text correcting the typos identified in lines 195 and 226 and in general trying to edit it properly to remove ambiguous statements.*

Fig1 panels H, I, and J and Fig2 panel F show Western Blots without the actin control

Response: *Corrected. We agree with the reviewer, we wanted to just show the ratio between phosphorylated and total protein expression in those panels. We thought that there was no necessary to show loading control (actin) as total and phosphorylated protein expression was shown for each lane but the reviewer is correct in that loading control should be shown as well.*